# Relax and Merge: A Simple Yet Effective Framework for Solving Fair $k$-Means and $k$-sparse Wasserstein Barycenter Problems

**Shihong Song , Guanlin Mo** [*]**, Hu Ding**[†]
School of Computer Science and Technology, University of Science and Technology of China
{shihongsong, moguanlin}@mail.ustc.edu.cn
huding@ustc.edu.cn

## Abstract

The fairness of clustering algorithms has gained widespread attention across various areas in machine learning. In this paper, we study *fair k-means clustering* in Euclidean space. Given a dataset comprising several groups, the fairness constraint requires that each cluster should contain a proportion of points from each group within specified lower and upper bounds. Due to these fairness constraints, determining the locations of $k$ centers and finding the induced partition are quite challenging tasks. We propose a novel "Relax and Merge" framework that returns a $(1 + 4\rho + O(\epsilon))$-approximate solution, where $\rho$ is the approximate ratio of an off-the-shelf vanilla $k$-means algorithm and $O(\epsilon)$ can be an arbitrarily small positive number. If equipped with a PTAS of $k$-means, our solution can achieve an approximation ratio of $(5 + O(\epsilon))$ with only a slight violation of the fairness constraints, which improves the current state-of-the-art approximation guarantee. Furthermore, using our framework, we can also obtain a $(1+4\rho+O(\epsilon))$-approximate solution for the $k$-sparse Wasserstein Barycenter problem, which is a fundamental optimization problem in the field of optimal transport, and a $(2 + 6\rho)$-approximate solution for the *strictly fair k-means clustering* with no violation, both of which are better than the current state-of-the-art methods. In addition, the empirical results demonstrate that our proposed algorithm can significantly outperform baseline approaches in terms of clustering cost.

## 1 Introduction

Clustering is one of the most fundamental problems in the area of machine learning. A wide range of practical applications rely on effective clustering algorithms, such as feature engineering (Glassman et al., 2014; Alelyani et al., 2018; Yuan et al., 2023; Zhang et al., 2023), image processing (Coleman & Andrews, 1979; Chang et al., 2017), and bioinformatics (Ronan et al., 2016; Nugent & Meila, 2010). In particular, the $k$-means clustering problem has been extensively studied in the past decades (Jain, 2010). Given an input dataset $P \subset \mathbb{R}^d$, the goal of the $k$-means problem is to find a set $S$ of at most $k$ points for minimizing the clustering cost, which is the sum of the squared distances from every point of $P$ to its nearest neighbor in $S$. In recent years, motivated by various fields like education, social security, and cultural communication, the study on *fairness* of clustering has in particular attracted a great amount of attention (Chierichetti et al., 2017; Bera et al., 2019; Huang et al., 2019; Chen et al., 2019; Ghadiri et al., 2021).

In this paper, we consider the problem of $(\alpha, \beta)$-***fair k-means clustering*** that was initially proposed by Chierichetti et al. (2017) and then generalized by Bera et al. (2019). Informally speaking, we assume that the given dataset $P$ consists of $m$ groups of points, and the "fairness" constraint requires that in each obtained cluster, the points from each group should take a fraction between pre-specified lower and upper bounds. Bera et al. (2019) showed that a $\rho$-approximate algorithm for vanilla

---

[*]The first two authors contribute equally.
[†]Corresponding author.

$k$-means can provide a $(2 + \sqrt{\rho})^2$- approximate solution [1] for $(\alpha, \beta)$-fair $k$-clustering with a slight violation on the fairness constraints, where the "violation" is formally defined in Section 2. Regarding the no violation scenario, Dai et al. (2022) and Wu et al. (2024) both obtained a $O(logk)$-approximate solution for fair $k$-median. Wu et al. (2024) achieved a quasi-polynomial-time approximate scheme. Furthermore, Böhm et al. (2021) studied the "strictly" fair $k$-means clustering problem, where it requires that the number of points from each group should be uniform in every cluster; they obtained a $(2 + \sqrt{\rho})^2$ approximate solution without violation. These fair $k$-means algorithms can also be accelerated by using the coreset techniques, such as (Huang et al., 2019; Braverman et al., 2022; Bandyapadhyay et al., 2024). There also exist polynomial-time approximation scheme (PTAS) for fair $k$-means, such as the algorithms proposed in (Böhm et al., 2021; Schmidt et al., 2020; Bandyapadhyay et al., 2024), but their methods have an exponential time complexity in $k$. We are also aware of several other different definitions of fairness for clustering problems, such as the *proportionally fair* clustering (Chen et al., 2019; Micha & Shah, 2020) and *socially fair $k$-means clustering* (Ghadiri et al., 2021; Abbasi et al., 2021; Makarychev & Vakilian, 2021; Chlamtáč et al., 2022).

Another problem closely related to fair $k$-means is the so-called "$k$-**sparse Wassertein Barycenter (WB)**" (Agueh & Carlier, 2011) (the formal definition is shown in Section 2). The Wasserstein Barycenter is a fundamental concept in optimal transport theory, and it represents the "average" or central distribution of a set of probability distributions. It plays a crucial role in various applications such as image processing (Bonneel et al., 2015; Cuturi & Doucet, 2014), data analysis (Rabin et al., 2012), and machine learning (Backhoff-Veraguas et al., 2022; Metelli et al., 2019). Given $m > 1$ discrete distributions, the goal of the $k$-sparse WB problem is to find a discrete distribution (i.e., the barycenter) that minimizes the sum of the Wasserstein distances (Villani, 2021) between itself to all the given distributions, and meanwhile the support size of the barycenter is restricted to be no larger than a given integer $k \geq 1$. If relaxing the "$k$-sparse" constraint (*i.e.,* the barycenter is allowed to take a support size larger than $k$), Altschuler & Boix-Adsera (2021) presented an algorithm based on linear programming, which can compute the WB within fixed dimensions in polynomial time. If the locations of the WB supports are given, the problem is called "fixed support WB", which can be solved by using several existing algorithms (Claici et al., 2018; Cuturi & Doucet, 2014; Cuturi & Peyré, 2016; Lin et al., 2020). If we keep the "$k$-sparse" constraint, it has been proved that the problem is NP-hard (Borgwardt & Patterson, 2021). To the best of our knowledge, the current lowest approximation ratio of k-sparse WB problem is also $(2 + \sqrt{\rho})^2$ (same with the aforementioned approximation factor for fair $k$-means), as recently studied by Yang & Ding (2024). In fact, we can regard this problem as a special case of fair $k$-means clustering, where each input distribution is an individual group and the unique cost measured by "Wasserstein distance" is implicitly endowed with a kind of fairness. This observation from Yang & Ding (2024) inspires us to consider solving the $k$-sparse WB problem under our framework.

**Why fair $k$-means is so challenging?** Though the fair $k$-means clustering has been extensively studied in recent years, their current state-of-the-art approximation qualities are still not that satisfying. The major difficulty arises from the lack of "locality property" (Ding & Xu, 2020; Bhattacharya et al., 2018) caused by fair constraints. More precisely, in a clustering result of vanilla $k$-means, each client point obviously belongs to its closest center. That is, a $k$-means clustering implicitly forms a *Voronoi diagram*, where the cell centers are exactly the $k$ cluster centers, and the client points in each Voronoi cell form a cluster. However, when we add some fair constraints, such as requiring that the proportion of points of each group should be equal in each cluster, the situation becomes more complicated. Given a set of cluster center locations, because the groups of client points within a Voronoi cell may not be equally distributed, some points are forced to be assigned to other Voronoi cells. This loss of locality introduces significant uncertainty for the selection of cluster center positions. The previous works (Bera et al., 2019; Böhm et al., 2021) do not pay much attention on how to handle this locality issue when searching for the cluster centers, instead, they directly apply vanilla $k$-means algorithms to the entire input dataset or a group, and use the obtained center locations as the center locations for fair $k$-means. It is easy to notice that their methods could result in a certain gap with the optimal fair $k$-means solution. To narrow this gap, we attempt to design some more effective way to determine the center locations, where the key part that we believe, should be how to encode the fair constraints into the searching algorithm.

---

[1] In their paper, the approximate ratio is written as $(2 + \rho)$ because they added a squared root to the $k$-means cost function.

**Our key ideas and main results.** Our key idea relies on an important observation, where we find that the fair $k$-means problem is inherently related to a classic geometric structure, "$\epsilon$-approximate centroid set", which was firstly proposed by Matoušek (2000). Roughly speaking, given a dataset, an $\epsilon$-approximate centroid set should contain at least one point that approximately represents the centroid location of any subset of this given dataset. It means that the $\epsilon$-approximate centroid set contains not only the approximate centroids based on the Voronoi diagram, but also the approximate centroids of those potential fairness-preserving clusters.

Inspired by the above observation, we illustrate the relationship between fair $k$-means and $\epsilon$-approximate centroid set first, and then propose a novel *Relax-and-Merge* framework. In this framework, we relax the constraints on the number of clusters $k$; we focus on utilizing fair constraints to cluster the data into small and fair clusters, which are then merged together to determine the positions of $k$ cluster centers. As shown in Table 1, our result is better than the state of the art works (Bera et al., 2019; Böhm et al., 2021). Equipped with a PTAS for $k$-means problem (e.g., the algorithm from Cohen-Addad et al. (2019)), our algorithm yields a $5 + O(\epsilon)$ approximation factor. We also present two important extensions from our work. The first extension is an $(1 + 4\rho + O(\epsilon))$ solution for k-sparse Wasserstein Barycenter. The second one is about strictly fair $k$-means. We give a refined algorithm of *Relax and Merge* that yields a no-violation solution with a $(2 + 6\rho)$ approximation factor, which is better than the state of the art work (Böhm et al., 2021).

| Algorithms | Approximation ratio | When $\rho = 1 + O(\epsilon)$ | Note on the quality |
|:---:|:---:|:---:|:---:|
| Bera et al. (2019) | $(2 + \sqrt{\rho})^2$ | $9 + O(\epsilon)$ | general case |
| Schmidt et al. (2020) | $5.5\rho + 1$ | $6.5 + O(\epsilon)$ | two groups only |
| Böhm et al. (2021) | $(2 + \sqrt{\rho})^2$ | $9 + O(\epsilon)$ | strictly only, no violation |
| Yang & Ding (2024) | $(2 + \sqrt{\rho})^2$ | $9 + O(\epsilon)$ | $k$-sparse WB |
| Algorithm 1, now | $1 + 4\rho + O(\epsilon)$ | $5 + O(\epsilon)$ | general case |
| Algorithm 2, now | $2 + 6\rho$ | $8 + O(\epsilon)$ | strictly only, no violation |

Table 1: Comparison of the approximation ratios for fair $k$-means and $k$-sparse WB. The "general case" includes $(\alpha, \beta)$-fair $k$-means, strictly $(\alpha, \beta)$-fair $k$-means and $k$-sparse WB.

**Other Related Works on $k$-Means** The vanilla $k$-means problem is a topic that has been widely studied in both theory and practice. It has been proved that $k$-means clustering is NP-hard even in $2D$ if $k$ is large (Mahajan et al., 2012). In high dimensions, even if $k$ is fixed, say $k = 2$, the $k$-means problem is still NP-hard (Drineas et al., 2004). Furthermore, Lee et al. (2017) proved the APX-hardness result for Euclidean $k$-means problem, which implies that it is impossible to approximate the optimal solution of $k$-means below a factor 1.0013 in polynomial time under the assumption of P $\neq$ NP. Therefore, a number of approximation algorithms have been proposed in theory. If the dimension $d$ is fixed, Kanungo et al. (2002) obtained a $(9 + O(\epsilon))$-approximate solution by using the local search technique. Roughly speaking, the idea of local search is swapping a small number of points in every iteration, so as to incrementally improve the solution until converging at some local optimum. Following this idea, Cohen-Addad et al. (2019) and Friggstad et al. (2019) proposed the PTASes for $k$-means in low dimensional space (or high dimensional space with constant doubling dimension). For high-dimensional case with constant $k$, Kumar et al. (2010) proposed an elegant peeling algorithm that iteratively finds the $k$ cluster centers and eventually obtain the PTAS. In addition, a simplified PTAS based on $D^2$ sampling was later proposed by Jaiswal et al. (2014).

## 2 PRELIMINARIES

**Notations.** In this paper, we always assume that the dimensionality $d$ of the Euclidean space is constant. Let $P$ denote the set of $n$ client points located in Euclidean space $\mathbb{R}^d$. The set $P$ consists of $m$ different groups (not necessarily disjoint), *i.e.*, $\boldsymbol{P} = \cup_{i=1}^m \boldsymbol{P^{(i)}}$, and each group has the size $|\boldsymbol{P^{(i)}}| = \boldsymbol{n^{(i)}}$ (we use the superscript "$(i)$" to denote the group's index). The Euclidean distance between two points $a, b \in \mathbb{R}^d$ is denoted by $\|a - b\|$; the distance between a point $a$ and any set $Q \subset \mathbb{R}^d$ is denoted by $\texttt{dist}(a, Q) = \min_{q \in Q} \|a - q\|$, and the nearest neighbor of $a$ in $Q$ is denoted as $\mathcal{N}(a, Q)$. The centroid of a set $Q$ is denoted by $\texttt{Cen}(Q)$.

For the vanilla $k$-means problem, the client points are always assigned to their nearest center. However, if the fairness constraint is considered, the assignment may not be that straightforward. To describe the fair $k$-means clustering more clearly, we introduce the "**assignment matrix**" first. Given any candidate set of $k$ cluster centers $S$, we define the assignment matrix $\phi_S : P \times S \to \mathbb{R}^+$ to indicate the assignment relation between the client points and cluster centers. For every $p \in P$ and $s \in S$, $\phi_S(p, s)$ denotes the proportion that is assigned to center $s$ (e.g., we may respectively assign 30% and 70% to two different centers). Obviously, we have $\sum_{s \in S} \phi_S(p, s) = 1$. For each center $s \in S$, we use $w(s) = \sum_{p \in P} \phi_S(p, s)$ to denote the amount of weight assigned to $s$; for each group $P^{(i)}$, we similarly define the function $w^{(i)}(s) = \sum_{p \in P^{(i)}} \phi_S(p, s)$. Let $\mathtt{Cost}(P, S, \phi_S)$ denote the cost of input instance $P$ with given $S$ and $\phi_S$:

$$\mathtt{Cost}(P, S, \phi_S) = \sum_{p \in P} \sum_{s \in S} \|p - s\|^2 \phi_S(p, s). \tag{1}$$

**Problem 1** (($\alpha, \beta$)-fair $k$-means clustering (Bera et al., 2019)). *Given an instance $P$ as described above and two parameter vectors $\alpha, \beta \in [0, 1]^m$, the goal of the ($\boldsymbol{\alpha}, \boldsymbol{\beta}$)-fair $k$-means clustering is to find the set $S$ consisting of $k$ points and an assignment matrix $\phi_S$, such that the clustering cost (1) is minimized, and meanwhile each cluster center $s \in S$ should satisfy the fairness constraint: $\beta_i w(s) \le w^{(i)}(s) \le \alpha_i w(s)$ for every $i \in \{1, 2, \cdots, m\}$. Here, we use $\alpha_i, \beta_i$ to denote the $i$-th entry of $\alpha$ and $\beta$, respectively.*

*Moreover, if the $m$ groups are disjoint with equal size (i.e., $n^{(i)} = n/m$ for any $i$), and $\alpha_i = \beta_i = 1/m$ for each group $P^{(i)}$, we say this is a **strictly fair $k$-means** clustering problem.*

For Problem 1, we can specify two types of solutions: **fractional** and **integral**. Their difference is only from the restriction on the assignment matrix $\phi_S$. For the first one, each entry $\phi_S(p, s)$ can be any real number between 0 and 1; but for the latter one, we require that the value of $\phi_S(p, s)$ should be either 0 or 1, that is, the whole weight of $p$ should be assigned to only one cluster center.

How to round a fractional solution into integral while preserving fairness constraints is still an open problem. Bera et al. (2019) introduced the **violation factor** to measure the violations of fairness constraints after rounding: an assignment matrix $\phi_S$ is a $\lambda$-violation solution if $\beta_i \sum_{p \in P} \phi_S(p, s) - \lambda \le \sum_{p \in P^{(i)}} \phi_S(p, s) \le \alpha_i \sum_{p \in P} \phi_S(p, s) + \lambda, \quad \forall s \in S, \forall i \in [m]$. In their paper, a fractional solution can always be rounded to integral, but it introduces some violations, which will be discussed in Section 3.1. In this paper, we use $OPT$ to denote the optimal integral cost of Problem 1. We use $S_{\mathtt{opt}} = \{\tilde{s}_1, \tilde{s}_2, \cdots, \tilde{s}_k\}$ to denote the optimal solution of integral fair $k$-means problem and its assignment matrix is denoted by $\phi_{S_{\mathtt{opt}}}$. For each $\tilde{s}_j$, let $C_j = \{p \in P \mid \phi_{S_{\mathtt{opt}}}(p, \tilde{s}_j) > 0\}$ be the corresponding cluster, i.e., the set of point assigned to it. A simple observation is that, if given a fixed candidate cluster centers set $S$, the assignment matrix $\phi_S$ can be obtained via solving a linear programming (we can view the $n \times k$ entries of $\phi_S$ as the variables):

$$\begin{aligned} \min_{\phi_S} \quad & \sum_{p \in P} \sum_{s \in S} \|p - s\|^2 \phi_S(p, s) \\ s.t. \quad & \beta_i \sum_{p \in P} \phi_S(p, s) \le \sum_{p \in P^{(i)}} \phi_S(p, s) \le \alpha_i \sum_{p \in P} \phi_S(p, s), \quad \forall s \in S, \forall i \in [m], \\ & \sum_{s \in S} \phi_S(p, s) = 1, \quad \forall p \in P. \end{aligned} \tag{2}$$

If we want to compute an integral solution, the above (2) should be an integer LP. Given a set $S$, $\phi_S^*$ denotes the optimal solution of (2) and $\tilde{\phi}_S$ denotes the corresponding optimal integral solution.

The following proposition is a folklore result that has been used in many articles on clustering algorithms (e.g., (Kanungo et al., 2002)). We will also repeatedly use it in our proofs.

**Proposition 1.** *Given a finite weighted point set $Q \subset \mathbb{R}^d$, for any point $a$, $\sum_{q \in Q} w(q)\|a - q\|^2 = \sum_{q \in Q} w(q)\|q - \mathtt{Cen}(Q)\|^2 + w(Q) \cdot \|a - \mathtt{Cen}(Q)\|^2$, where $w(Q)$ is the total weight of $Q$.*

Next we introduce an important geometric structure "$\epsilon$-approximate centroid set", which was firstly proposed by Matoušek (2000). Roughly speaking, the $\epsilon$-approximate centroid set approximately covers the centroids of any subset of given data, even though the subsets do not align with the "Voronoi diagram" structure.

**Definition 1.** *Given a finite set $P \subset \mathbb{R}^d$ and a small parameter $\epsilon > 0$, we use $\mathtt{CS}_\epsilon(P)$ to denote an $\epsilon$-approximate centroid set of $P$ that satisfies: for any nonempty subset $Q \subseteq P$, there always exists a point $v \in \mathtt{CS}_\epsilon(P)$ such that $\|v - \mathtt{Cen}(Q)\| \le \frac{\epsilon}{3} \sqrt{\frac{1}{|Q|} \sum_{q \in Q} \|q - \mathtt{Cen}(Q)\|^2}$.*

**Remark 1.** *Matoušek (2000) also presented a construction algorithm based on the space partitioning technique "quadtree" (Finkel & Bentley, 1974). In Appendix A, we briefly illustrate the role of the $\epsilon$-approximate centroid set in preserving fairness constraints and how to construct it. The size of the obtained $\epsilon$-approximate centroid set is $O(|P|\epsilon^{-d} \log(1/\epsilon))$ and the construction time complexity is $O(|P| \log |P| + |P|\epsilon^{-d} \log(1/\epsilon))$.*

Next, we give the formal definition of $k$**-sparse Wasserstein Barycenter** problem.

**Definition 2** (Wasserstein Distance). *Let $P$ and $Q$ be weighted point sets supported in $\mathbb{R}^d$. Wasserstein distance is the minimum transportation cost between $P$ and $Q$: $\mathcal{W}(P,Q) = \min_F \sqrt{\sum_{p \in P} \sum_{q \in Q} F(p,q) \|p - q\|^2}$, where the transport matrix $F : P \times Q \to [0,1]$ should satisfy: $\sum_{p \in P} F(p,q) = w(q)$ for any $q \in Q$, and $\sum_{q \in Q} F(p,q) = w(p)$ for any $p \in P$, where $w(p)$ (or $w(q)$) is the weight of point $p$ (or $q$) such that $\sum_{p \in P} w(p) = \sum_{q \in Q} w(q)$.*

For a weighted set $S$, we use $\mathtt{supp}(S)$ to denote its support, i.e., the set that shares the same location of $S$ but not weighted. The number of points is $\mathtt{supp}(S)$ is denoted by $|\mathtt{supp}(S)|$.

**Problem 2** ($k$-sparse Wassertein Barycenter ($k$-sparse WB)). *Given $m$ discrete probability distributions $P^{(1)}, \cdots, P^{(m)}$ supported on $\mathbb{R}^d$, WB is the probability distribution $S$ minimizing the sum of squared Wasserstein distances to them, i.e., $\arg\min_S \sum_{i=1}^m \mathcal{W}^2(P^{(i)}, S)$. The problem is called $k$-**sparse Wasserstein Barycenter** if we restrict $|\mathtt{supp}(S)| \le k$*

In Section 3.2, we explain why this problem can be regarded as a fair $k$-means clustering.

# 3  OUR "RELAX AND MERGE" FRAMEWORK

In general, there are two stages in clustering with fair constraints. The first stage is to find the proper locations of clustering centers, and the second stage is to assign all the client points to the centers by solving LP (2). The previous approaches often use the vanilla $k$-means in the first stage to obtain the location of centers, and then take the fairness into account in the second stage (Bera et al., 2019; Böhm et al., 2021). In our proposed framework, we aim to shift the consideration of fair constraints to the first stage, so as to achieve a lower approximation factor in the final result. The following theorem is our main result.

**Theorem 1.** *Given an instance of Problem 1 and a $\rho$-approximate vanilla $k$-means clustering algorithm, there exists an algorithm that can return a fractional $(1 + 4\rho + O(\epsilon))$ approximate solution for Problem 1. Further, one can apply a rounding method to transform this fractional solution to an integral one with a constant violation factor while ensuring the cost does not increase.*

The details for computing the fractional solution are shown in Algorithm 1. The set $T$ in the relax stage contains the approximate centroids of all the potential clusters with preserving fair constraints. Then we solve a linear program to obtain the relaxed solution $(T, \phi_T^*)$ that also preserves the fair constraints. Because of that, the merge stage is able to determine the appropriate locations for the cluster centers of Problem 1.

## 3.1  DETAILS FOR $(\alpha, \beta)$ FAIR $k$-MEANS ALGORITHM

First, we mainly focus on the fractional version of $(\alpha, \beta)$-fair $k$-means problem. More precisely, we allow the value of the assignment function $\phi_S$ to be a real number in $[0,1]$ rather than $\{0,1\}$. To prove Theorem 1, we need the following lemmas first. Specifically, Lemma 1 provides the bound for the cost from the merged solution $S$; Lemma 2 shows that the $\epsilon$-approximate centroid set provides a satisfied relaxed solution with a cost no more than $(1 + O(\epsilon))OPT$. Combining with the rounding methods, Theorem 1 can be obtained.

**Lemma 1.** *Let $\eta$ be any positive number. If we suppose $\mathtt{Cost}(P, T, \phi_T^*) \le \eta \cdot OPT$, then the solution $(S, \phi_S^*)$ returned by Algorithm 1 is an $(\eta + (2\eta + 2)\rho)$-approximate solution for Problem 1.*

---

**Algorithm 1:** FRACTIONAL FAIR $k$-MEANS

---

**Input:** The dataset $P$, $k$, $\alpha$, $\beta$, and $\epsilon > 0$

1 **Relax**: Construct a relaxed solution $T$, *i.e.*, an $\epsilon$-approximate centroid set, such that
$\text{Cost}(P, T, \phi_T^*) \leq (1 + O(\epsilon)) \cdot OPT$ (see Lemma 2). Here, we relax the size constraint of
centers to be polynomial of $n$ rather than exactly $k$, so as to achieve a sufficiently low cost.

2 Solve LP (2) on $T$ to obtain the optimal assignment matrix $\phi_T^*$. $T$ and $\phi_T^*$ can be viewed as a
relaxed solution for $(\alpha, \beta)$-fair $k$-means, *i.e.*, the number of centers may be more than $k$, and
meanwhile, the cost is bounded and the fairness constraints are also preserved.

3 Adjust $T$. For each $t \in T$, we update the location of $t$ to be the corresponding cluster centroid
$\pi(t) = \frac{\sum_{p \in P} p \cdot \phi_T^*(p,t)}{w(t)}$. The adjusted $T$ is denoted by $\pi(T)$.

4 **Merge:** Run a $\rho$-approximate $k$-means algorithm on $\pi(T)$ to obtain centers set $S$. Then, solve
LP (2) on $S$ to obtain the optimal assignment matrix $\phi_S^*$.

5 **return** $S$ *and* $\phi_S^*$

---

*Proof.* According to the definition of fractional fair $k$-means problem, the cost can be written as

$$\text{Cost}(P, S, \phi_S^*) = \sum_{p \in P} \sum_{s \in S} \|p - s\|^2 \phi_S^*(p, s). \tag{3}$$

Now we consider another assignment strategy: we firstly assign $P$ to $T$ according to $\phi_T^*$ ( recall that
$\phi_T^*$ is the optimal fractional assignment matrix from $P$ to $T$), and then we assign every weighted
point in $T$ to some $s \in S$ such that $s$ is closest point to $\pi(t)$. Since $\phi_S^*$ is the optimal assignment
matrix from $P$ to $S$, the cost of this assignment strategy should have:

$$\sum_{p \in P} \sum_{t \in T} \|p - \mathcal{N}(\pi(t), S)\|^2 \phi_T^*(p, t) \geq \text{Cost}(P, S, \phi_S^*). \tag{4}$$

Since $\pi(t)$ is the centroid of the weighted points assigned to $t$, according to Proposition 1, we know
the left-hand side of (4) should have the upper bound

$$\sum_{t \in T} \Big[ \sum_{p \in P} \|p - \pi(t)\|^2 \phi_T^*(p, t) + \|\pi(t) - \mathcal{N}(\pi(t), S)\|^2 w(t) \Big]$$
$$= \underbrace{\sum_{p \in P} \sum_{t \in T} \|p - \pi(t)\|^2 \phi_T^*(p, t)}_{(a)} + \underbrace{\sum_{t \in T} \|\pi(t) - \mathcal{N}(\pi(t), S)\|^2 w(t)}_{(b)}. \tag{5}$$

Then we bound (a) and (b) separately.

$$(a) = \sum_{p \in P} \sum_{t \in T} \|p - \pi(t)\|^2 \phi_T^*(p, t) \leq \sum_{p \in P} \sum_{t \in T} \|p - t\|^2 \phi_T^*(p, t) \leq \eta \cdot OPT. \tag{6}$$

The first inequality holds because $\pi(t)$ is the centroid of the weighted points assigned to $t$, minimizing
the weighted sum of the squared distances between them. The second inequality holds because
$\text{Cost}(P, T, \phi_T^*) \leq \eta \cdot OPT$. Next, we focus on (b). Suppose $S_{means}$ is the optimal $k$-means solution
of $\pi(T)$. Then we have:

$$(b) = \sum_{t \in T} \|\pi(t) - \mathcal{N}(\pi(t), S)\|^2 w(t) \leq \rho \sum_{t \in T} \|\pi(t) - \mathcal{N}(\pi(t), S_{means})\|^2 w(t)$$
$$= \rho \sum_{p \in P} \sum_{t \in T} \|\pi(t) - \mathcal{N}(\pi(t), S_{means})\|^2 \phi_T^*(p, t)$$
$$= \rho \sum_{p \in P} \sum_{t \in T} \Big[ \sum_{\tilde{s} \in S_{opt}} \|\pi(t) - \mathcal{N}(\pi(t), S_{means})\|^2 \phi_{S_{opt}}^*(p, \tilde{s}) \Big] \phi_T^*(p, t)$$
$$\leq \rho \sum_{p \in P} \sum_{t \in T} \Big[ \sum_{\tilde{s} \in S_{opt}} \|\pi(t) - \tilde{s}\|^2 \phi_{S_{opt}}^*(p, \tilde{s}) \Big] \phi_T^*(p, t). \tag{7}$$

The first inequality holds because of the $\rho$-approximate $k$-means algorithm. The second equality holds because for every $t \in T$, $\sum_{p \in P} \phi_T^*(p, t) = w(t)$. The third equality holds because for every $p \in P$, $\sum_{\tilde{s} \in S_{opt}} \phi_{S_{opt}}^*(p, \tilde{s}) = 1$. And the last equality holds because of the optimality of $S_{means}$. Further, according to squared triangle inequality, we have

$$
\begin{aligned}
(b) &\leq \rho \sum_{p \in P} \sum_{t \in T} \Big[ \sum_{\tilde{s} \in S_{opt}} \big[ \|\pi(t) - p\| + \|p - \tilde{s}\| \big]^2 \phi_{S_{opt}}^*(p, \tilde{s}) \Big] \phi_T^*(p, t) \\
&\leq \rho \sum_{p \in P} \sum_{t \in T} \sum_{\tilde{s} \in S_{opt}} 2\|\pi(t) - p\|^2 \phi_{S_{opt}}^*(p, \tilde{s}) \phi_T^*(p, t) \\
&\quad + \rho \sum_{p \in P} \sum_{t \in T} \sum_{\tilde{s} \in S_{opt}} 2\|p - \tilde{s}\|^2 \phi_{S_{opt}}^*(p, \tilde{s}) \phi_T^*(p, t) \\
&= 2\rho \sum_{p \in P} \sum_{t \in T} \|\pi(t) - p\|^2 \phi_T^*(p, t) + 2\rho \sum_{p \in P} \sum_{\tilde{s} \in S_{opt}} \|p - \tilde{s}\|^2 \phi_{S_{opt}}^*(p, \tilde{s}).
\end{aligned}
\tag{8}
$$

The last equality holds because for any $p \in P$, $\sum_{\tilde{s} \in S_{opt}} \phi_{S_{opt}}^*(p, \tilde{s}) = 1$. The first term is exactly $2\rho$ times of (a) and the second term equals $2\rho \cdot OPT$. Through combining (a) and (b), we can obtain an approximation factor of $\eta + (2\eta + 2)\rho$. $\qquad\square$

Algorithm 1 reduces the fair $k$-means problem to computing the set $T$. The following lemma shows that an $\epsilon$-approximate centroid set is a good candidate for $T$.

**Lemma 2.** *If $T$ is an $\epsilon$-approximate centroid set of $P$, then $\mathtt{Cost}(P, T, \phi_T^*) \leq (1 + O(\epsilon))OPT$.*

*Proof.* According to Definition 1, let $t_j \in T$ denote the point such that $\|t_j - \mathtt{Cen}(C_j)\| \leq \frac{\epsilon}{3} \sqrt{\frac{1}{|C_j|} \sum_{p \in C_j} \|p - \mathtt{Cen}(C_j)\|^2}$, where $C_j$ is the $j$-th optimal cluster. Let $T' = \{t_1, \cdots, t_k\}$. A key observation is that each optimal center $\tilde{s}_j$ is always the centroid of $C_j$, *i.e.*, $\mathtt{cen}(C_j) = \tilde{s}_j$, so we have $\|t_j - \mathtt{Cen}(C_j)\|^2 \leq \frac{\epsilon^2}{9|C_j|} \sum_{p \in C_j} \|p - \tilde{s}_j\|^2 = \frac{\epsilon^2}{9|C_j|} OPT_j$, where $OPT_j = \sum_{p \in C_j} \|p - \tilde{s}_j\|^2$.

If we assign all points of $C_j$ to $t_j$, the cost of every $C_j$ can be written as $\sum_{p \in C_j} \|t_j - p\|^2 =$

$$
\sum_{p \in C_j} \|t_j - \tilde{s}_j\|^2 + \sum_{p \in C_j} \|p - \tilde{s}_j\|^2 \leq \frac{\epsilon^2}{9} OPT_j + OPT_j = (1 + O(\epsilon))OPT_j.
\tag{9}
$$

The first equality holds due to Proposition 1. Since $\phi_{T'}^*$ is the optimal assignment matrix of $T'$, $\mathtt{Cost}(P, T', \phi_{T'}^*) \leq \sum_{j=1}^k \sum_{p \in C_j} \|t_j - p\|^2 \leq (1 + O(\epsilon)) \sum_{j=1}^k OPT_j = (1 + O(\epsilon))OPT$. Finally, since $T'$ is a subset of $T$, we have $\mathtt{Cost}(P, T, \phi_T^*) \leq \mathtt{Cost}(P, T', \phi_{T'}^*) \leq (1 + O(\epsilon))OPT$. $\qquad\square$

Through combining Lemma 1 and Lemma 2, we can immediately obtain Lemma 3.

**Lemma 3.** *Equipped with the $\epsilon$-approximate centroid set by Matoušek (2000), the cost of the solution returned by Algorithm 1 is at most $(1 + 4\rho + O(\epsilon))OPT$. Furthermore, by utilizing the PTAS of vanilla $k$-means algorithm, the cost of the solution is at most $(5 + O(\epsilon))OPT$.*

**Rounding for integral solution.** Note that Lemma 3 only guarantees a fractional solution. Recall the "violation factor" introduced in Section 2. According to the rounding method proposed in (Bera et al., 2019), a fractional solution of Problem 1 can be rounded to be integral with $(3\Delta + 4)$ violation, where $\Delta$ is the maximum number of groups a point can join in (e.g., if a point can belong to three groups, $\Delta$ should be equal to 3). Their main idea is to reduce the fair assignment problem to the *minimum degree-bounded matroid basis* (MBDMB) problem, and then solve the MBDMB by iteratively solving a linear program (LP). In the current article, we further propose a new rounding method that can improve this violation factor to "2" when assuming $\Delta = 1$, *i.e.*, the groups are mutually **disjoint**, meaning that each point belongs to exactly one group (if using the method of (Bera et al., 2019), the factor should be 7). Actually, it is natural to assume that the groups are disjoint, e.g., each person may belong to one race. Fair clustering problem in disjoint groups has also been studied in Bercea et al. (2019); Wu et al. (2024); Chierichetti et al. (2017). Our key idea is building a **"hub-guided" minimum cost circulation** problem. Roughly speaking, we utilize a set of carefully designed "hubs"

in a transportation network, for guiding the integral fair matching between the input points and the obtained cluster centers. We show the result in Lemma 4, and place the proof to Appendix C.

**Lemma 4.** *If the groups are mutually disjoint, one can round the fractional solution returned by Algorithm 1 to be integral with at most 2-violation, while the cost does not increase.*

Finally, Theorem 1 can be obtained by combining either the rounding method from (Bera et al., 2019) for general case, or Lemma 4 for disjoint case.

**Overall time complexity.** As we mentioned in Remark 1, computing an $\epsilon$-approximate centroid set of $P$ needs $O(n \log n + n\epsilon^{-d} \log(1/\epsilon))$ time. Suppose the time complexities of linear programming, vanilla $k$-means are denoted by $\mathcal{T}_{LP}$ and $\mathcal{T}_{means}$, respectively. The time complexity of the adjustment of $T$ is $O(|T|n)$, which is dominated by $\mathcal{T}_{LP}$ because the size of $\phi_T^*$ is $n \times |T|$. Therefore, the overall time complexity of Algorithm 1 is $O(n \log n + n\epsilon^{-d} \log(1/\epsilon)) + \mathcal{T}_{LP} + \mathcal{T}_{means}$. It is worth noting the the complexity can be further reduced by using the assignment preserving coreset ideas (Huang et al., 2019; Braverman et al., 2022; Bandyapadhyay et al., 2024). By doing this, we need to introduce an extra running time for coreset construction, which is linear to $n$, but we can compress the data size from $n$ to $poly(k, \epsilon)$.

## 3.2 EXTENSION TO $k$-SPARSE WASSERSTEIN BARYCENTER

A cute property of Algorithm 1 is that it can be easily extended to address the $k$-sparse WB problem. Recall the definition of $k$-sparse WB in Problem 2. The given $m$ distributions can be viewed as $m$ groups of weighted points. And the sum of Wasserstein distances between barycenter and given distributions can be rewritten as the sum of squared Euclidean distances from $P$ to the centers. Moreover, the flows induced by Wasserstein distances between barycenter and the given distributions can implicitly ensure the fairness, *i.e.*, for each point $s$ in barycenter, $w^{(i)}(s) = \frac{1}{m}w(s)$ for any $i \in [m]$. Namely, we can directly perform our "Relax and Merge" framework by setting $\alpha_i = \beta_i = 1/m$. First, we calculate the $\epsilon$-approximate centroid (here we ignore the weight of points) set to obtain $T$, then we use $T$ as the support of the Barycenter to run a "fixed support" WB algorithm (Claici et al., 2018; Cuturi & Doucet, 2014; Cuturi & Peyré, 2016; Lin et al., 2020) to obtain the weights of $T$ (due to the space limit, we leave some details on fixed support WB algorithms to Appendix D). Finally, we run a vanilla $k$-means algorithm on $T$ to obtain the $k$-sparse solution.

**Theorem 2.** *If $T$ is an $\epsilon$-approximate centroid set of $\cup_{i=1}^{m} P^{(i)}$, Algorithm 1 returns a $(1+4\rho+O(\epsilon))$-approximate solution for $k$-sparse Wasserstein Barycenter problem.*

## 3.3 STRICTLY FAIR $k$-MEANS WITHOUT VIOLATION

Since the strictly fair $k$-means is a special case of $(\alpha, \beta)$-fair $k$-means, by using Algorithm 1 and the rounding technique introduced by Section 3.1, we can obtain an integral solution but with certain violation. In this section, we consider how to obtain an integral solution with no violation. Specifically, we compute the fairlet decomposition (Chierichetti et al., 2017) for the input groups and use its centroids as the relaxed solution $T$ rather than $\epsilon$-approxiamte centroid set. First, we give the definition of fairlet decomposition for multiple groups, which extends the original definition of (Chierichetti et al., 2017) from two groups to multiple groups.

**Definition 3** (Fairlet Decomposition). *Given a dataset $P$ that has $m$ equal-sized disjoint groups, We say a set $G$ of $m$ points is a fairlet of $P$, if $G$ contains exactly one point from each group of $P$. A set $\mathcal{G}$ of $n/m$ fairlets is a fairlet decomposition of $P$, if all fairlets in $\mathcal{G}$ are disjoint, where $n/m$ is the number of points in each group of $P$.*

We define the cost of fairlet decomposition $\mathcal{G}$ as $\texttt{Cost}_{\texttt{fairlet}}(\mathcal{G}) = \sum_{G \in \mathcal{G}} \sum_{p \in G} \|p - \texttt{Cen}(G)\|^2$. It is easy to know that fairlet decomposition is indeed a solution of strictly fair $n/m$-means. Hence, we can still use the "Relax and Merge" technique: regard the centroids of fairlets in fairlet decomposition as a relaxed solution, and then run $\rho$-approximate vanilla $k$-means algorithm on these centroids. So, we reduce the strictly fair $k$-means problem to the fairlet decomposition problem. We propose Algorithm 2, which first computes a 2-approximate fairlet decomposition and then generates a $(2+6\rho)$-approximate integral solution for strictly fair $k$-means.

**Theorem 3.** *Algorithm 2 returns a $(2+6\rho)$-approximate integral solution of strictly fair $k$-means.*

To prove Theorem 3, we need to prove the following lemma, which shows that $\mathcal{G}$ is a 2-approximate fairlet decomposition. Then, we can use the same idea of Lemma 1 to obtain Theorem 3. Recall

---

**Algorithm 2:** STRICYTLY FAIR $k$-MEANS

---

**Input:** The dataset $P = \cup_{i=1}^m P^{(i)}$, $k$

1 **for** $i = 1$ *to* $m$ **do**

2     **for** $j = 1$ *to* $m$ *and* $i \neq j$ **do**

3         Compute the perfect one-to-one matching $\tau_{ij}$ between $P^{(i)}$ and $P^{(j)}$ (pairwise matching cost is the squared distance) by using the Hungarian algorithm (Kuhn, 1955). For each point $p \in P^{(i)}$, the point matched with $p$ in $P^{(j)}$ is denoted as $\tau_{ij}(p)$.

4     **end**

5     Construct a fairlet decomposition $\mathcal{G}_i$ (initially empty) according to the matchings: for each point $p \in P^{(i)}$, add the fairlet $\{\tau_{i1}(p), \tau_{i2}(p), \cdots, \tau_{im}(p)\}$ to $\mathcal{G}_i$.

6 **end**

7 Choose $\mathcal{G}_v$ where $v = \arg\min_i \sum_{p \in P^{(i)}} \sum_{j=1}^m \|p - \tau_{ij}(p)\|^2$ as $\mathcal{G}$.

8 Construct the relaxed solution $T = \{\texttt{Cen}(G) \mid G \text{ is any fairlet of } \mathcal{G}\}$.

9 Run a $\rho$-approximate $k$-means algorithm on $T$, and obtain the solution $S$.

10 **Integral assignment:** Assign all the points according to the fairlet decomposition $\mathcal{G}$, *i.e.*, if a point $p$ belongs to some fairlet $G$, then assign $p$ to $\mathcal{N}(\texttt{Cen}(G), S)$.

11 **return** *S and the obtained integral assignment*

---

that Lemma 1 shows that if we have a relaxed solution $T$ with a bounded cost $\eta \cdot OPT$, then the merged solution will have constant approximate ratio. Here, $T$ obtained by Algorithm 2 also provides a relaxed solution whose cost does not exceed $2OPT$. Hence, after we merge $T$ and obtain $S$, the approximate ratio should no more than $(\eta + (2\eta + 2)\rho) = 2 + 6\rho$. Furthermore, if we use PTAS for $k$-means, the overall approximate ratio of Algorithm 2 is $8 + O(\epsilon)$.

**Lemma 5.** *If $\mathcal{G}$ is the fairlet decomposition obtained by Algorithm 2, then $\texttt{Cost}_{fairlet}(\mathcal{G}) \leq 2OPT$.*

*Proof.* Suppose $G$ is a fairlet, and we use $G^{(i)}$ to denote the point in $G$ and belongs to group $P^{(i)}$, *i.e.*, $G^{(i)}$ is the unique point of $G \cap P^{(i)}$. Now we bound the $\texttt{Cost}_{fairlet}(\mathcal{G})$ as follows:

$$
\begin{aligned}
\texttt{Cost}_{fairlet}(\mathcal{G}) &= \sum_{G \in \mathcal{G}} \sum_{p \in G} \|p - \texttt{Cen}(G)\|^2 \\
&\leq \sum_{G \in \mathcal{G}} \sum_{p \in G} \|p - \texttt{Cen}(G)\|^2 + m \sum_{G \in \mathcal{G}} \|G^{(v)} - \texttt{Cen}(G)\|^2,
\end{aligned}
\tag{10}
$$

where $v$ is the "best" group index selected in Line 7 in Algorithm 2. We use $\mathcal{G}_{OPT}$ to denote the optimal fairlet decomposition that has the lowest cost (we cannot obtain $\mathcal{G}_{OPT}$ in reality, and here we just use it for conducting our analysis). For each $p \in P$, let $\mathcal{G}_{OPT}(p)$ denote the fairlet of $\mathcal{G}_{OPT}$ that $p$ belongs to, *i.e.*, $p \in \mathcal{G}_{OPT}(p)$. Suppose that $P^{(u)}$ is the "closest" group to $\mathcal{G}_{OPT}$, *i.e.* $u = \arg\min_{i \in [m]} \sum_{p \in P^{(i)}} \|\texttt{Cen}(\mathcal{G}_{OPT}(p)) - p\|^2$. According to Proposition 1, the right side of (10) equals to $\sum_{p \in P^{(v)}} \sum_{j=1}^m \|p - \tau_{vj}(p)\|^2$, so we have $\texttt{Cost}_{fairlet}(\mathcal{G}) \leq$

$$
\sum_{p \in P^{(v)}} \sum_{j=1}^m \|p - \tau_{vj}(p)\|^2 \leq \sum_{p \in P^{(u)}} \sum_{j=1}^m \|p - \tau_{uj}(p)\|^2 \leq \sum_{p \in P^{(u)}} \sum_{j=1}^m \|p - (\mathcal{G}_{OPT}(p))^{(j)}\|^2. \tag{11}
$$

The first inequality holds because $v = \arg\min_i \sum_{p \in P^{(i)}} \sum_{j=1}^m \|p - \tau_{ij}(p)\|^2$. And the last inequality holds because $\tau$ is the perfect one-to-one matching. Using Proposition 1 again, we have $\texttt{Cost}_{fairlet}(\mathcal{G}) \leq$

$$
\sum_{p \in P^{(u)}} \sum_{j=1}^m \|\texttt{Cen}(\mathcal{G}_{OPT}(p)) - (\mathcal{G}_{OPT}(p))^{(j)}\|^2 + m \sum_{p \in P^{(u)}} \|\texttt{Cen}(\mathcal{G}_{OPT}(p)) - p\|^2. \tag{12}
$$

Note that $\mathcal{G}_{OPT}$ is the optimal fairlet decomposition, as well as the optimal strictly fair $n/m$-means solution, so the first term of (12) should be at most $OPT$. As for the second term, since $P^{(u)}$ is the "closest" group to $\mathcal{G}$, it should be no larger than $m \cdot \frac{1}{m} OPT \leq OPT$ (because the minimum distance "$\sum_{p \in P^{(u)}} \|\texttt{Cen}(\mathcal{G}_{OPT}(p)) - p\|^2$" should not exceed the average distance $\frac{1}{m} OPT$). Overall, we complete the proof of Lemma 5. $\square$

## 4 EXPERIMENTS

In this section, we perform the empirical evaluation on our algorithms. Our experiments are conducted on a server equipped with Intel(R) Xeon(R) Gold 6154 CPU @ 3.00GHz CPU and 512GB memory. We implement our algorithms in C++ and python (with linear programming solver gurobi (Gurobi Optimization, LLC, 2023)). We use the following datasets which are commonly used in previous works: **Bank** (Moro et al., 2014)(4522 points with 5 groups), **Adult** (Becker & Kohavi, 1996) (32561 points with 7 groups), **Census** (Zhou & Chen, 2002)(50000 points with 10 groups), **creditcard** (Yeh & Lien, 2009) (30000 points with 8 groups), **Biodeg** (Mansouri et al., 2013) (1055 points with 2 groups), **Breastcancer** (Wolberg,William, Mangasarian,Olvi, Street,Nick, and Street,W., 1995) (570 points with 2 groups), **Moons** (scikit-learn developers, 2007-2023) (200 points with 2 groups), **Hypercube**(200 points with 2 groups), **Cluto** (Karypis et al., 1999) (800 points with 8 groups), and **Complex** (800 points with 8 groups). The last four datasets consist of disjoint and equal sized groups, so we can perform strictly fair $k$-means algorithms on them. We place the detailed information of these datasets in Appendix F. Regarding the selection of $\alpha$ and $\beta$, we set $\alpha_i = \beta_i = \frac{|P^{(i)}|}{|P|}$ and we also discuss more choices for $\alpha$ and $\beta$, and provide more experimental results, including the part of $k$-sparse Wasserstein Barycenter, in the Section F of the appendix. We use $k$-means++ (Arthur & Vassilvitskii, 2007) as the $k$-means solver in our Algorithm 1.

**Results on $(\alpha, \beta)$-Fair $k$-means.** We compared the cost of $(\alpha, \beta)$-fair $k$-means of our Algorithm 1 and baselines. We choose the algorithm proposed by Bera et al. (2019) (denoted by NIPS19) and Böhm et al. (2021) (denoted by ORL21) as the baselines. The construction of an $\epsilon$-approximate centroid set is a theoretical algorithm that can be replaced by some efficient methods in practice. In our experiments, we adopted the alternative implementation by Kanungo et al. (2002), which combines the kd-tree (Friedman et al., 1977) and a sampling technique. Figure 1 shows that our algorithm gives the lowest cost of $(\alpha, \beta)$-fair $k$-means, indicating that Algorithm 1 can find better center locations. This improvement is possible due to that our method considers the fairness information of groups when choosing the locations of centers.

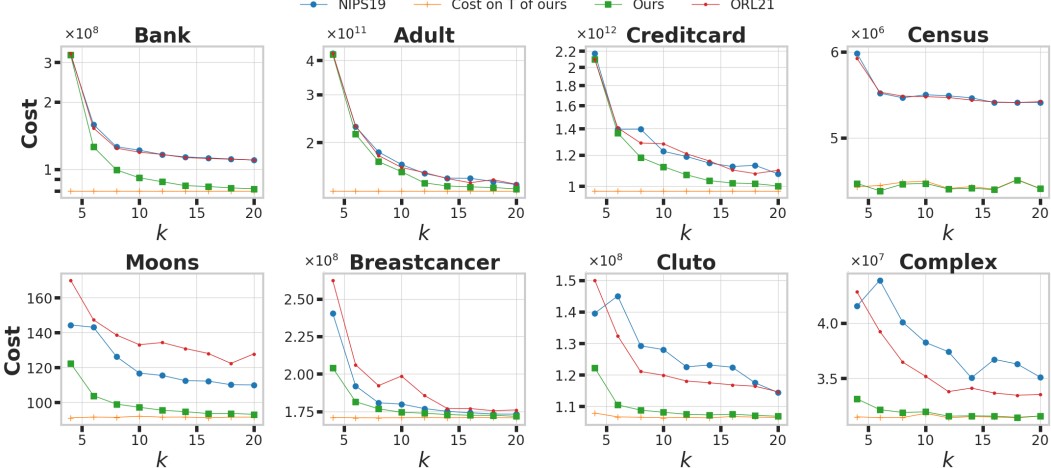

Figure 1: The cost obtained by the algorithms with different $k$.

## 5 CONCLUSION

In this paper, we utilize the insight on the relationship between the fair $k$-means problem and a classic geometric structure, $\epsilon$-approximate centroid set, for developing a novel "Relax and Merge" framework. It can achieve a $(1 + 4\rho + O(\epsilon))$ approximation ratio of fair $k$-means and $k$-sparse Wasserstein Barycenter problems, which improves the current state-of-the-art approximation guarantees. There still exists some open problems: how to obtain an integral approximate solution of general case without violation? In addition, is it possible to extend our 'Relax and Merge" framework to other types of clustering problems, such as the proportionally fair clustering (Chen et al., 2019) and socially fair $k$-means clustering (Ghadiri et al., 2021)?

## 6 ACKNOWLEDGEMENT

The research of this work was supported in part by the National Key Research and Development Program of China (NO.2021YFA1000900), the National Natural Science Foundation of China (NO.62272432, NO.62432016) and the Natural Science Foundation of Anhui Province (NO.2208085MF163). We want to thank the anonymous reviewers for their helpful comments. We are grateful to Prof. Lingxiao Huang in Nanjing University and Qingyuan Yang for inspiring discussions.

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

## A ϵ-APPROXIMATE CENTROID SET

The algorithm of constructing an $\epsilon$-approximate centroid set is proposed by Matoušek (2000). Here we briefly introduce the idea. First, we use a quadtree to partition the space into hierarchical cubes. At each level of the tree, we construct a grid. The length of the grid is set to ensure that the grid points can always cover all approximate centroids of all cubes at this level. The approximate centroid set is the union of all grid points across all levels.

In Figure 2, we visually illustrate the difference between the $k$-means clustering center and the fair $k$-means clustering center. The vanilla $k$-means induces a Voronoi diagram, so that every $k$-means center is located at the centroid of a $k$-means cluster. However, a fair $k$-means center can be located at the centroid of any potential cluster that satisfies the fairness constraints. The $\epsilon$-approximate centroid set structure can help us to find these potential centroids and preserves the fairness constraints for the later procedures.

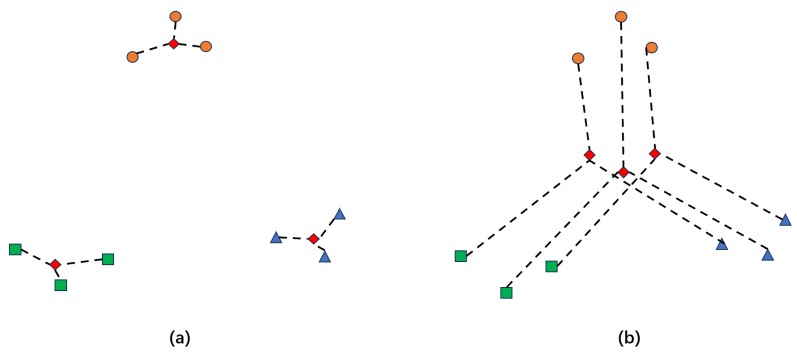

Figure 2: The difference between the location of $k$-means clustering centers and the fair $k$-means clustering centers. The input dataset contains 3 different groups represented by orange, blue, and green points respectively. The red diamonds represent the cluster centers under different assumptions for the clustering problem. (a) shows the clustering result of $k$-means, while (b) shows the clustering result of fair $k$-means.

## B OMITTED PROOFS

**Theorem 2** If $T$ is an $\epsilon$-approximate centroid set of $\cup_{i=1}^{m} P^{(i)}$, Algorithm 1 returns a $(1+4\rho+O(\epsilon))$-approximate solution for $k$-sparse Wasserstein Barycenter problem.

To prove this theorem, we need the following lemma.

**Lemma 6.** *If $T$ is an $\epsilon$-approximate centroid set of $\cup_{i=1}^{m} P^{(i)}$ and $w(t)$ for each $t \in T$ is obtained by solving LP(2), then $T$ is a $(1 + O(\epsilon))$-approximate Wasserstein Barycenter.*

*Proof.* A critical fact is that there exist an optimal Wasserstein Barycenter $T^*$ such that all points of $T^*$ located in the centroid of some fairlet of $P$. This claim has been proved in (Anderes et al., 2016) (Section 2, Equation 4). Therefore, if we calculate an $\epsilon$-approximate centroid set $T$, then $T$ can always cover the locations of $T^*$, *i.e.*, $\mathtt{Cost}(P, T, \phi_T^*) \leq (1 + O(\epsilon))\mathtt{Cost}(P, T^*, \phi_{T^*}^*) \leq (1 + O(\epsilon))OPT$. So using the same proof idea with Lemma 2, we can obtain the conclusion of Lemma 6. □

Combine Lemma 6 and Lemma 1, we arrive at Theorem 2.

## C THE ROUNDING TECHNIQUE

Our rounding algorithm consists of three steps: constructing a network structure of Minimum Cost Circulation Problem (MCCP), setting the parameters of each edge based on a fractional solution obtained by Algorithm 1, and solving the MCCP above. This reduction to MCCP is inspired by

Ding & Xu (2015) (Section 4.3), while having some fundamental differences with their method. Our algorithm has different objectives compared to theirs, as it is based on a different approach to setting network parameters, and our method offers better time complexity guarantees. Our rounding algorithm requires only a single call to the minimum-cost circulation algorithm, and it can be completed in $O(n^3k^2)$ time even when using the vanilla Edmonds-Karp algorithm (Dinitz, 1970; Edmonds & Karp, 1972).

The process of our algorithm is described as follows. Recall that the dataset $P$ consists of $m$ different groups, *i.e.*, $P = \cup_{i=1}^{m} P^{(i)}$ and we assume that the groups are disjoint. By executing the Algorithm 1, we obtain a center set $S$ and corresponding fractional assignment matrix $\phi_S^*$. Now, in order to build a minimum cost circulation instance, we need to construct a network structure as Figure 3 and for each arc $(u, v)$, we should set the lower/upper bound of the flow $f(u, v)$ and its cost $c(u, v)$. We create a copy of $S$, denoted by $S^{(i)}$, for each group $P^{(i)}$. Each $S^{(i)}$ is a **"hub"** used for transit, specifically to receive weights from group $P^{(i)}$ and transmit them to $S$. To facilitate understanding, we can imagine that each $s_l^{(i)} \in S^{(i)}$, where $l \in [k]$, and its corresponding $s_l \in S$ is in the same position, but only accepts the weights from group $P^{(i)}$. We set $c(p_j^{(i)}, s_l^{(i)})$, *i.e.*, the cost of the arc from any $p_j^{(i)} \in P^{(i)}$, where $j \in [n^{(i)}]$, to an $s_l^{(i)} \in S^{(i)}$ to be $||p_j^{(i)} - s_l^{(i)}||^2$. The cost of the remaining arcs are 0.

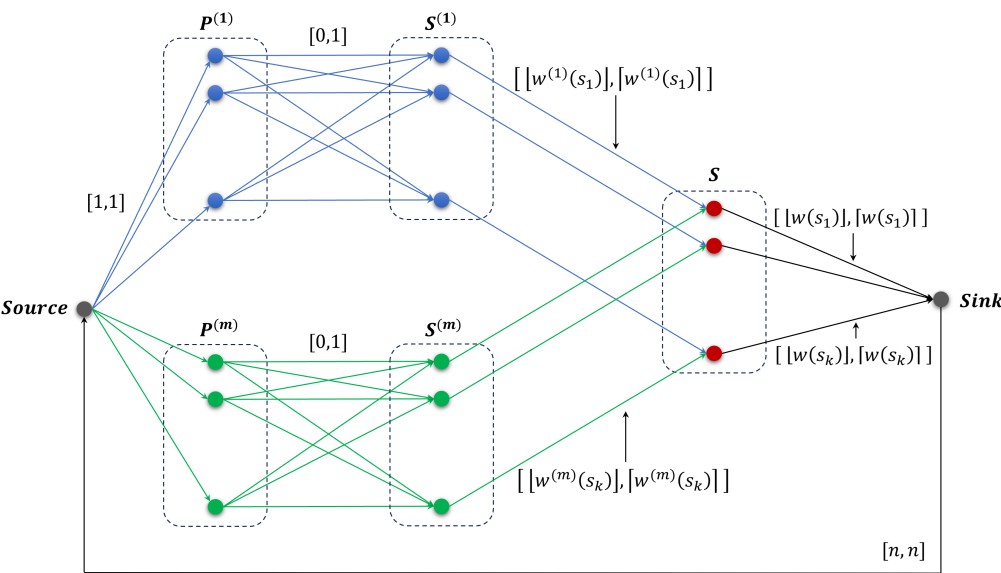

Figure 3: The instance of the minimum cost circulation problem established through $(S, \phi_S^*)$. The upper and lower bounds of the flow for each arc are annotated in the graph.

Next, we set the lower bound and the upper bound of the flow on each arc, as shown in Figure 3. First, the flow from the "Source" node to each $p \in P$ is restricted to 1, which means that each point $p \in P^{(1)}$ has a weight of 1 to assign to $S^{(1)}$. Then, between each $P^{(i)}$ and its "hub" $S^{(i)}$, the flow from each $p_j^{(i)} \in P^{(i)}$ to each $s_l^{(i)} \in S^{(i)}$ is bounded by $[0, 1]$. Here, the flow $f(p_j^{(i)}, s_l^{(i)})$ denotes the amount of the weight that assigned from $p_j^{(i)}$ to $s_l \in S$ in an $(\alpha, \beta)$-fair $k$-means solution. Subsequently, recall that in the solution $(S, \phi_S^*)$ we obtained before, the weight received by a center $s_l \in S$ from group $P^{(i)}$ is $w^{(i)}(s_l)$. We bound the flow $f(s_l^{(i)}, s_l)$ by $\left[ \lfloor w^{(i)}(s_l) \rfloor, \lceil w^{(i)}(s_l) \rceil \right]$. Finally, the flow from each $s_l \in S$ to the "Sink" node is bounded by $\left[ \lfloor w(s_l) \rfloor, \lceil w(s_l) \rceil \right]$, and we set $f(Sink, Source) = n$ to form a circulation. At this point, we have established an instance of the minimum cost circulation problem, denoted by $MCCP(S, \phi_S^*)$. Obviously, we have the following observation:

**Observation 1.** $\phi_S^*$ *induces a feasible solution of* $MCCP(S, \phi_S^*)$.

The observation is straightforward because the flow induced by $\phi_S^*$ meet all the bounds applied to the flow. Then, we give the proof of Lemma 4 mentioned in Section 3.1.

**Lemma 4.** There exists an algorithm that can round a fractional solution of $(\alpha, \beta)$-fair $k$-means to integral with at most 2-violation while the cost does not increase.

*Proof.* It is known that the minimum cost circulation problem has an integrality property (Cormen et al., 2009), which guarantees that if the arcs have integer capacities, there will always be an optimal solution with integer flow values on each arc. Utilizing an algorithm for minimum cost circulation problem or minimum cost flow problem (the two problems are equivalent), which converges to an integer solution like Ford-Fulkerson (Ford & Fulkerson, 1956), we can obtain an integer optimal solution of $MCCP(S, \phi_S^*)$, which has a cost no larger than the solution induced by $\phi_S^*$.

Next, we prove that the assignment matrix, say $\phi_S'$, induced by the integer optimal solution of $MCCP(S, \phi_S^*)$ is a 2-violation assignment from $P$ to $S$. Recall that we presented the definition of the violation factor in Section 2: An assignment matrix $\phi_S$ is a $\lambda$-violation solution if $\beta_i \sum_{p \in P^{(i)}} \phi_S(p, s) - \lambda \leq \sum_{p \in P^{(i)}} \phi_S(p, s) \leq \alpha_i \sum_{p \in P} \phi_S(p, s) + \lambda, \quad \forall s \in S, \forall i \in [m]$. According to the construction procedure of $MCCP(S, \phi_S^*)$, the lower bound of the flow $f(s_l^{(i)}, s_l)$ is $\lfloor w^{(i)}(s_l) \rfloor$, which satisfies:

$$
\begin{aligned}
\lfloor w^{(i)}(s_l) \rfloor &\geq \lfloor \alpha^{(i)}(\lceil w(s_l) \rceil - 1) \rfloor \\
&= \lfloor \alpha_i \lceil w(s_l) \rceil - \alpha_i \rfloor \\
&\geq \lceil \alpha_i \lceil w(s_l) \rceil - \alpha_i \rceil - 1 \\
&\geq (\alpha_i \lceil w(s_l) \rceil - \alpha_i) - 1.
\end{aligned}
\tag{13}
$$

Note that the upper bound of the flow $f(s_l, Sink)$ is $\lceil w(s_l) \rceil$ so we have:

$$
\begin{aligned}
\lfloor w^{(i)}(s_l) \rfloor &\geq \alpha_i \lceil w(s_l) \rceil - \alpha_i - 1 \\
&\geq \alpha_i \lceil w(s_l) \rceil - 2,
\end{aligned}
\tag{14}
$$

and similarly,

$$
\begin{aligned}
\lceil w^{(i)}(s_l) \rceil &\leq \beta_i \lfloor w(s_l) \rfloor + \beta_i + 1 \\
&\leq \beta_i \lfloor w(s_l) \rfloor + 2,
\end{aligned}
\tag{15}
$$

which indicates that $\phi_S'$ is a 2-violation assignment and complete the proof of Lemma 4. □

## D  FIXED SUPPORT WASSERSTEIN BARYCENTER

Given $m$ discrete distributions (weighted point sets, each set has total weight sum to 1) $P^{(1)}, \cdots, P^{(m)}$ and a set $T$ of WB, the objective of fixed support WB as follows:

$$
\begin{aligned}
\min_x \quad & \frac{1}{m} \sum_{l=1}^{m} \sum_{i=1}^{n^{(i)}} \sum_{j=1}^{n^{(j)}} \| P_i^{(l)} - T_j \|^2 x_{ij}^{(l)} \\
s.t. \quad & \sum_{j=1}^{|T|} x_{ij}^{(l)} = 1, \quad \forall l \in [m], \forall i \in [n^{(l)}] \\
& \sum_{i=1}^{n^{(l)}} x_{ij}^{(l)} w(P_i^{(w)}) = y_j, \quad \forall l \in [m], \forall j \in [|T|] \\
& \sum_{j=1}^{|T|} y_j = 1, \\
& x_{ij}^{(l)} \geq 0, \quad \forall l \in [m], \forall i \in [n^{(l)}], \forall j \in [|T|] \\
& y_j \geq 0, \quad \forall j \in [|T|]
\end{aligned}
\tag{16}
$$

It is easy to see that fixed support WB problem can be solved using linear programming method. Several existing works on solving LP (16) include (Claici et al., 2018; Cuturi & Doucet, 2014; Cuturi & Peyré, 2016; Lin et al., 2020).

For the sake of completeness, we need to clarify how the solution to the $k$-sparse Wasserstein barycenter solution is guaranteed to be a distribution. After we run Algorithm 1, we obtain the support $S$ (the locations of centers) of the returned solution and the assignment matrix $\phi_S^*$ (the transportation weight from $p = P_i^{(l)}$ to $f = S_j$ is denoted by $\phi_S^*(p, f) = x_{ij}^{(l)}$ in (16)). The key question is how to ensure that the summation of the weight of points in $S$ is equal to 1. Let us consider an arbitrary given distribution (or "group" in the context of fair $k$-means), e.g., $P^{(l)}$. For every facility $f$ in $S$, we define its weight $w(f) = \sum_{p \in P^{(l)}} \phi_S^*(p, f)$. This ensures that the total weight of $S$ must be equal to the total weight of $P^{(l)}$, which is 1 because $P^{(l)}$ is a distribution. The choice of $P^{(l)}$ can be arbitrary because, recall that $k$-sparse WB can be seen as a special fractional version of strictly fair $k$-means, meaning no matter which given distribution you choose, you will obtain the same weight distribution of $S$. The optimization will not change by setting the weight of $S$ because the weight of $S$ does not affect the cost.

# E    EXTEND ALGORITHM 1 TO $k$-MEDIAN AND $k$-MEANS IN GENERAL METRIC SPACE

Although we mainly consider the fair $k$-means problem in Euclidean space in this paper, for the sake of completeness, in this section, we illustrate how to extend our framework to solve $k$-median and $k$-means in general metric space. In summary, if the potential facility set is given, our framework achieves a $(1 + 2\rho)$-approximate solution for $k$-median ($(2 + 8\rho)$-approximate solution for $k$-means) in metric space, where $\rho$ is the approximation ratio for vanilla $k$-median ($k$-means) with a constant violation factor. If the metric space has a fixed doubling dimension (Gupta et al., 2003), then equipped with existing PTAS for metric $k$-median and $k$-means (Cohen-Addad et al., 2021; 2019; Friggstad et al., 2019), the best approximation ratios our framework can achieve are $(3 + O(\epsilon))$ for fair $k$-median and $(10 + O(\epsilon))$ for fair $k$-means.

Unfortunately, our theoretical guarantees in general metric space are weaker than those of Bera et al. (2019), in which they obtained a $(\rho + 2)$-approximation for $k$-median and a $(\sqrt{\rho} + 2)^2$-approximation for $k$-means. The obstacle to achieving a better approximation ratio for our framework is the "candidate set". In Euclidean space, we have an approximate centroid set. However, in general metric space, how can we obtain a candidate set that has similar properties to Proposition 1, which provides a more powerful tool than the basic triangle inequality? This is not only a potential future work of our framework but also an important open theoretical problem.

$k$-**Median in metric space.**    Firstly, we consider fair $k$-median in general metic space. We use $\texttt{dist}(\cdot, \cdot)$ to denote the distance between two points. We assume that the potential facility set $T$ is given. Therefore, in Algorithm 1, we just use the given facility set $T$ rather than computing the approximate centroid set. The cost of fair $k$-median can be written as

$$\texttt{Cost}(P, S, \phi_S^*) = \sum_{p \in P} \sum_{s \in S} \texttt{dist}(p, s) \phi_S^*(p, s). \tag{17}$$

Similar to Lemma 1, we have the following lemma.

**Lemma 7.** *Let $\eta$ be any positive number. If we suppose $\texttt{Cost}(P, T, \phi_T^*) \leq \eta \cdot OPT$, then the solution $(S, \phi_S^*)$ returned by Algorithm 1 (the construction of $T$ should be slightly changed) is an $(\eta + (\eta + 1)\rho)$-approximate solution for fair $k$-median problem in metric space, where $\rho$ is the approximation ratio of vanilla $k$-median.*

*Proof.* Now we consider another assignment strategy: we firstly assign $P$ to $T$ according to $\phi_T^*$ ( recall that $\phi_T^*$ is the optimal fractional assignment matrix from $P$ to $T$), and then we assign every weighted point in $T$ to some $s \in S$ such that $s$ is closest point to $\pi(t)$. Since $\phi_S^*$ is the optimal

assignment matrix from $P$ to $S$, the cost of this assignment strategy should have:

$$
\begin{aligned}
\texttt{Cost}(P, S, \phi_S^*) &\leq \sum_{p \in P} \sum_{t \in T} \texttt{dist}(p, \mathcal{N}(t, S)) \phi_T^*(p, t) \\
&\leq \sum_{t \in T} \sum_{p \in P} \Big[ \texttt{dist}(p, t) + \texttt{dist}(t, \mathcal{N}(t, S)) \Big] \phi_T^*(p, t) \\
&= \underbrace{\sum_{p \in P} \sum_{t \in T} \texttt{dist}(p, t) \phi_T^*(p, t)}_{(a)} + \underbrace{\sum_{p \in P} \sum_{t \in T} \texttt{dist}(t, \mathcal{N}(t, S)) \phi_T^*(p, t)}_{(b)}.
\end{aligned}
\tag{18}
$$

The second inequality is triangle inequality. Then we bound (a) and (b) separately. Firstly,

$$
(a) = \sum_{p \in P} \sum_{t \in T} \texttt{dist}(p, t) \phi_T^*(p, t) = \texttt{Cost}(P, T, \phi_T^*) \leq \eta \cdot OPT
\tag{19}
$$

Next, we focus on (b). Suppose $S_{median}$ is the optimal $k$-median solution of $T$. Then we have:

$$
\begin{aligned}
(b) &= \sum_{p \in P} \sum_{t \in T} \texttt{dist}(t, \mathcal{N}(t, S)) \phi_T^*(p, t) \\
&\leq \rho \sum_{p \in P} \sum_{t \in T} \texttt{dist}(t, \mathcal{N}(t, S_{median}) \phi_T^*(p, t) \\
&= \rho \sum_{p \in P} \sum_{t \in T} \Big[ \sum_{\tilde{s} \in S_{opt}} \texttt{dist}(t, \mathcal{N}(t, S_{median})) \phi_{S_{opt}}^*(p, \tilde{s}) \Big] \phi_T^*(p, t) \\
&\leq \rho \sum_{p \in P} \sum_{t \in T} \Big[ \sum_{\tilde{s} \in S_{opt}} \texttt{dist}(t, \tilde{s}) \phi_{S_{opt}}^*(p, \tilde{s}) \Big] \phi_T^*(p, t).
\end{aligned}
\tag{20}
$$

Further, according to the triangle inequality, we have

$$
\begin{aligned}
(b) &\leq \rho \sum_{p \in P} \sum_{t \in T} \Big[ \sum_{\tilde{s} \in S_{opt}} \big[ \texttt{dist}(t, p) + \texttt{dist}(p, \tilde{s}) \big] \phi_{S_{opt}}^*(p, \tilde{s}) \Big] \phi_T^*(p, t) \\
&\leq \rho \sum_{p \in P} \sum_{t \in T} \sum_{\tilde{s} \in S_{opt}} \texttt{dist}(t, p) \phi_{S_{opt}}^*(p, \tilde{s}) \phi_T^*(p, t) \\
&\quad + \rho \sum_{p \in P} \sum_{t \in T} \sum_{\tilde{s} \in S_{opt}} \texttt{dist}(p, \tilde{s}) \phi_{S_{opt}}^*(p, \tilde{s}) \phi_T^*(p, t) \\
&= \rho \sum_{p \in P} \sum_{t \in T} \texttt{dist}(t, p) \phi_T^*(p, t) + \rho \sum_{p \in P} \sum_{\tilde{s} \in S_{opt}} \texttt{dist}(p, \tilde{s}) \phi_{S_{opt}}^*(p, \tilde{s}).
\end{aligned}
\tag{21}
$$

The last equality holds because for any $p \in P$, $\sum_{\tilde{s} \in S_{opt}} \phi_{S_{opt}}^*(p, \tilde{s}) = 1$ and $\sum_{\tilde{t} \in T} \phi_T^*(p, t) = 1$. The first term is exactly $\rho$ times of (a) and the second term equals $\rho \cdot OPT$. Through combining (a) and (b), we can obtain an approximation factor of $\eta + (\eta + 1)\rho$. $\qquad \square$

$k$**-Means in metric space.** Using the same idea of Lemma 7 with squared triangle inequality $\texttt{dist}^2(a, b) \leq 2\texttt{dist}^2(a, c) + 2\texttt{dist}^2(c, b)$, we can immediately obtain the following corollary.

**Corollary 1.** *Let $\eta$ be any positive number. If we suppose $\texttt{Cost}(P, T, \phi_T^*) \leq \eta \cdot OPT$, then the solution $(S, \phi_S^*)$ returned by Algorithm 1 (slightly changed as above) is an $\big(2\eta + (4\eta + 4)\rho\big)$-approximate solution for fair $k$-means problem in metric space, where $\rho$ is the approximation ratio of vanilla $k$-means.*

When considering $k$-clustering problem in metric space, we usually assume that the potential facility set is given. We just use it as our candidate set $T$. Hence, the $\eta = 1$ in the above analysis, which leads a $(2 + \rho)$-approximation for fair $k$-median and a $(2 + 8\rho)$-approximation for fair $k$-means.

# F SUPPLEMENTARY EXPERIMENT

## F.1 DATASETS

The detailed information of our datasets is shown in Table 2. The group partition of every dataset is based on the "Group Column". Every group column has some group values. The set of groups is the Cartesian product of group values of all group column. For example, the groups of **Bank** dataset are (married, yes), (married, no), (single, yes), (single, no), (divorced, yes), (divorced, no). For large dataset **Census** and **Creditcard**, we sample 1000 points to make sure the LP solver works in acceptable time.

| Dataset | Size | Dimension | Group Column | Groups Values |
|---|---|---|---|---|
| Bank | 9999 | 3 | marital | married, single, divorced |
| | | | default | yes, no |
| Adult | 4522 | 5 | sex | female, male |
| | | | race | Amer-ind, asian-pac-isl, black, other, white |
| Creditcard | 30000 | 5 | marriage | married, single, other, null |
| | | | education | 7 groups |
| Census1990 | 50000 | 12 | dAge | 8 groups |
| | | | iSex | female, male |
| Moons | 200 | 2 | color | 2 groups |
| Hypercube | 200 | 3 | color | 2 groups |
| Complex | 3032 | 2 | color | 9 groups |
| Cluto | 10000 | 2 | color | 8 groups |
| Breastcancer | 570 | 31 | label | 2 groups |
| Biodeg | 1055 | 40 | label | 2 groups |

Table 2: Detailed Datasets Information

## F.2 COMPARISON ON COST OF STRICTLY FAIR $k$-MEANS

We compare our strictly fair $k$-means algorithm with the state-of-the-art algorithm ORL21 (Böhm et al., 2021). Both ORL21 and Algorithm 2 can return integral solution with no violation. Figure 4 shows that our method has significant advantage in terms of the clustering cost.

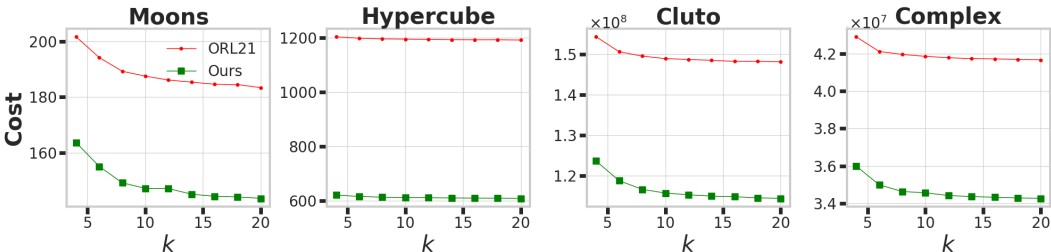

Figure 4: The cost of strictly fair $k$-means.

## F.3 COMPARISON ON COST WITH DIFFERENT $k$ AND $(\alpha, \beta)$

In the main paper, we set $\alpha_i = \beta_i = \frac{|P^{(i)}|}{|P|}$. Here, we try different $\alpha$ and $\beta$ to compare our algorithm to baselines. In order to make sure that the values of $\alpha$ and $\beta$ are feasible, we introduce the parameter $\delta \in (0, 1)$, which represents the degree of relaxation of fairness constraints, with a larger $\delta$ indicating looser constraints. We set $\alpha_i = \frac{|P^{(i)}|}{|P|} \cdot \frac{1}{1-\delta}$ and $\beta_i = \frac{|P^{(i)}|}{|P|} \cdot (1 - \delta)$. We set $\delta = 0.1$ and $0.2$ to compare the cost with baselines. The results are shown in Figure 5 and Figure 6, respectively.

In fact, as $\delta$ increases, the fairness constraints of the $(\alpha, \beta)$-fair $k$-means problem become more relaxed, and the corresponding fair $k$-means problem approaches the vanilla $k$-means problem. In cases where $\delta$ is large, in each cluster, the legal range of points from each group is larger, making the protection of fairness constraints less important, thus resulting in the optimal fair $k$-means center positions being very close to the centers of vanilla $k$-means. In the Table 2 of (Böhm et al., 2021), it is mentioned that when $\delta = 0.2$, the clustering results of vanilla $k$-means only violate the fairness constraints by 0.4%-2%, which makes our algorithm less advantageous under a relatively relaxed $\delta$ value.

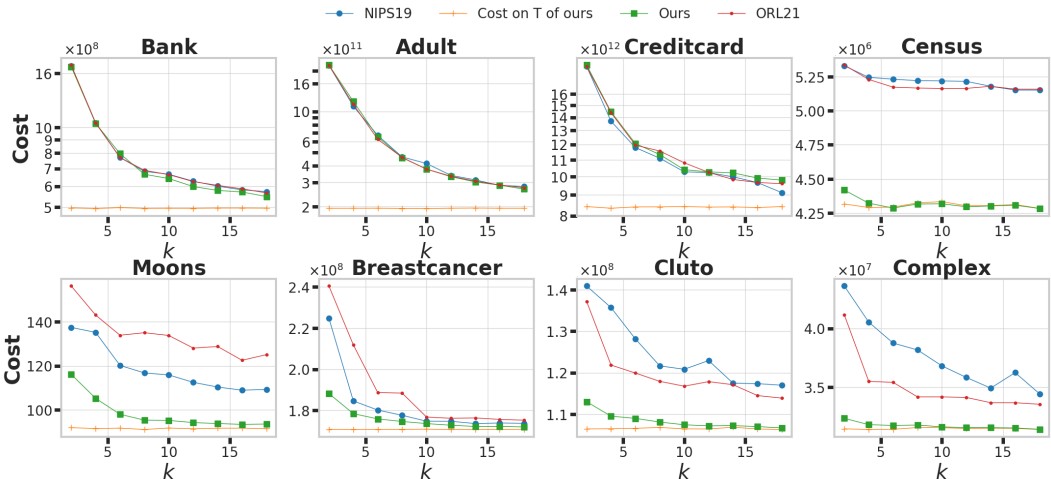

Figure 5: Comparison on Clustering Cost with $\delta = 0.1$

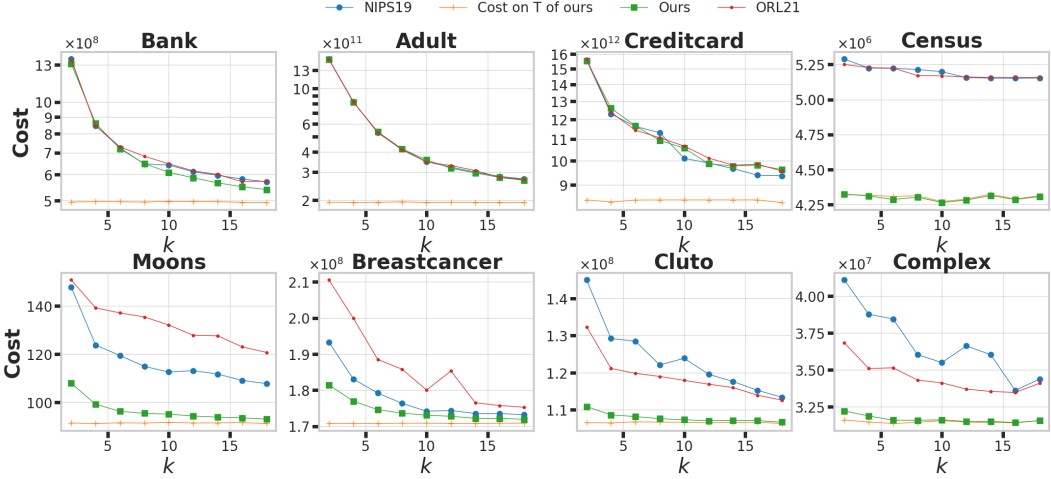

Figure 6: Comparison on Clustering Cost with $\delta = 0.2$

### F.4 COMPARISON ON COST OF $k$-SPARSE WASSERSTEIN BARYCENTER

We compare our algorithm with the very recent work (Yang & Ding, 2024) (denoted by IJCAI24) who obtain $(2 + \sqrt{\rho})^2$-approximate solution of $k$-sparse WB. The results are shown in Figure 7. In most cases, our algorithm can achieve a 10%-30% cost advantage over the previous work.

### F.5 COST ON DIFFERENT SAMPLING RATIO

In our algorithm, the most time consuming step is to solve LP(2) on $T$. A key observation during our experiment is that, after solving LP(2) on $T$, a large amount of points of $T$ have weight of 0.

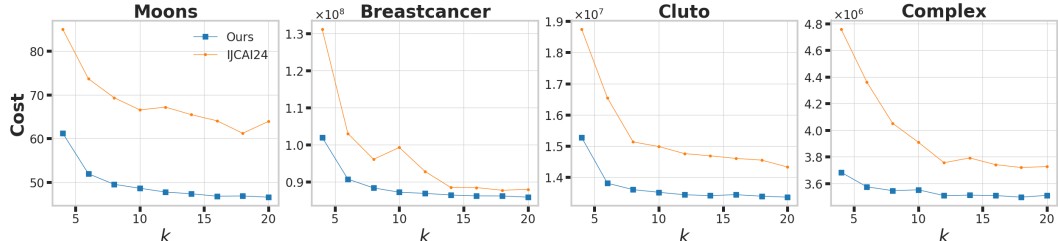

Figure 7: Comparison on the Cost of $k$-sparse Wasserstein Barycenter

Therefore, it is possible to reduce the size of $T$ while maintain the quality of $T$. Meanwhile, smaller $T$ helps to reduce the running time. In order to verify our thoughts, we use sampling method after we obtain $T$. We use sampling ratio of $100\%$, $50\%$, $20\%$ and $10\%$ and calculate the final cost of Algorithm 1 with different $k$. The results are shown in Figure 891011. In these figures, we can see that in most cases, the cost of sampled $T$ do not increase too much ($50\%$ sample yields no more than $10\%$ cost increasing and even $10\%$ sample yields no more than $20\%$ cost increasing in most cases).

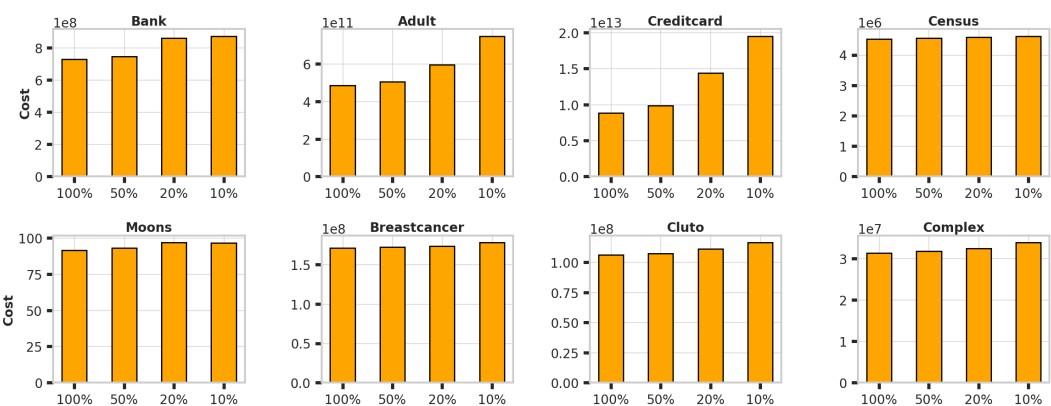

Figure 8: The cost on centriod set $T$ with different sampling ratio when $k = 5$

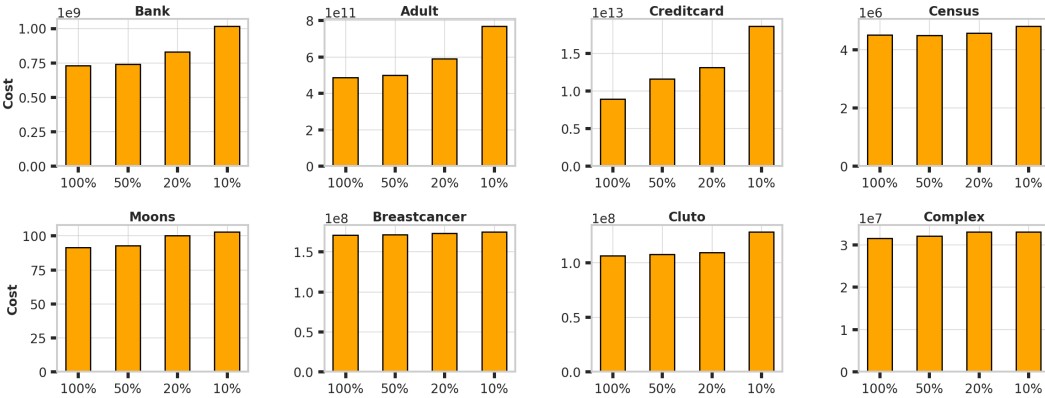

Figure 9: The cost on centriod set $T$ with different sampling ratio when $k = 10$

### F.6   RUNNING TIME WITH DIFFERENT SAMPLING RATIO ON $T$

As we discussed in F.5, sampling on relaxed solution $T$ can reduce the running time while the overall cost not increasing too much. We also test the running time with different sampling ratio. In summary, the running time of solving LP(2) on $T$ and overall Algorithm 1, shown in Table 3 and Table 4, can be significantly reduced by sampling.

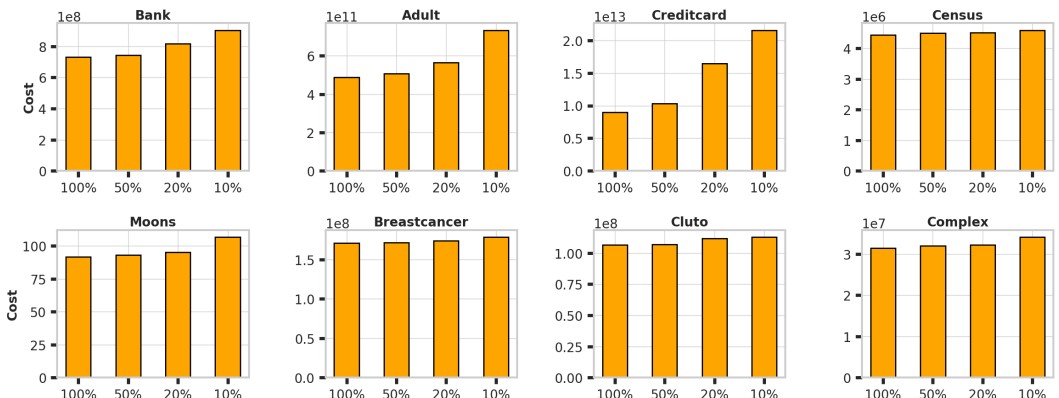

Figure 10: The cost on centriod set $T$ with different sampling ratio when $k = 15$

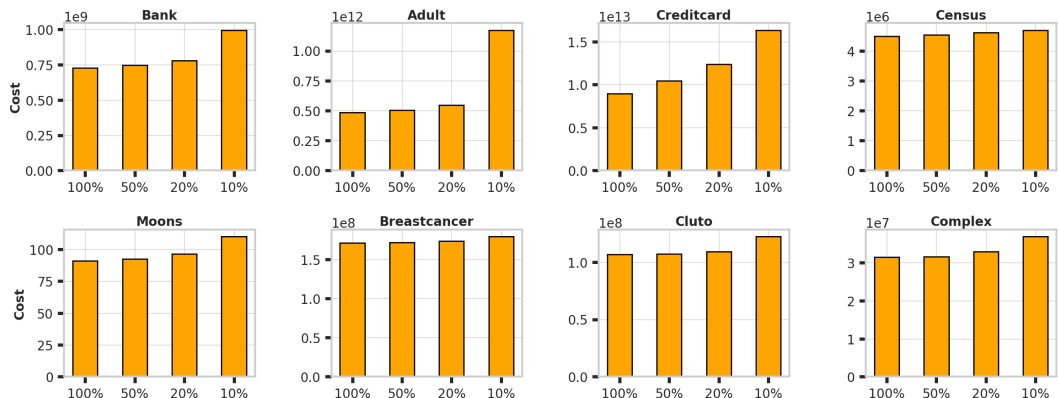

Figure 11: The cost on centriod set $T$ with different sampling ratio when $k = 20$

| Dataset | 100% | 50% | 20% | 10% |
|---|---|---|---|---|
| Bank | 39.97 | 19.14 | 7.12 | 3.39 |
| Adult | 66.48 | 28.58 | 9.67 | 4.64 |
| Creditcard | 80.235 | 32.51 | 11.08 | 5.43 |
| Census | 76.46 | 37.78 | 13.96 | 6.64 |
| Moons | 3.75 | 1.89 | 0.68 | 0.33 |
| Breastcancer | 11.03 | 5.28 | 2.01 | 1.07 |
| Cluto | 192.57 | 91.72 | 36.03 | 18.18 |
| Complex | 49.70 | 24.74 | 9.11 | 4.54 |

Table 3: Time (seconds) of solving LP(2) on $T$ with different sampling ratio

| Dataset | 100% | 50% | 20% | 10% |
|---|---|---|---|---|
| Bank | 42.03 | 21.20 | 9.16 | 5.44 |
| Adult | 69.19 | 31.24 | 12.31 | 7.34 |
| Creditcard | 83.42 | 35.62 | 14.20 | 8.57 |
| Census | 80.23 | 41.62 | 17.86 | 10.48 |
| Moons | 4.07 | 2.15 | 0.97 | 0.60 |
| Breastcancer | 11.78 | 6.05 | 2.68 | 1.67 |
| Cluto | 201.54 | 100.64 | 45.82 | 27.74 |
| Complex | 52.23 | 27.25 | 11.66 | 7.05 |

Table 4: Overall time (seconds) with different sampling ratio of $T$ when $k = 20$

### F.7 COMPARISON OF RUNNING TIME WITH BASELINES

We compared the running time of our algorithm (Algorithm 1 with our rounding technique) with the baseline NIPS19 (Bera et al., 2019). For strictly fair datasets, we also tested the running time of Algorithm 2 and ORL21 (Böhm et al., 2021). The results are shown in Table 5 and Table 6. Below, we provide a detailed analysis on the comparisons.

**Comparison between Algorithm 1 and NIPS19 (Bera et al., 2019).** Algorithm 1 and NIPS19 both have two important subprocedures: linear programming and the $k$-means algorithm. These two steps are the bottlenecks for Algorithm 1 and NIPS19. Specifically, NIPS19 first runs the $k$-means algorithm (*i.e.*, $k$-means++), and then calls the LP solver once to compute the fractional assignment. A different part of our Algorithm 1 is that it calls the LP solver twice, once to compute the weights of candidate set $T$ and once to compute the fractional assignment, and calls the $k$-means algorithm once. In Algorithm 1, we only need to run $k$-means on $T$, which should be much smaller than the whole dataset, leading to less running time for the $k$-means subprocedure compared to NIPS19. However, the first call to the LP solver to compute the weight of $T$ consumes more time than the second call because $|T| > k$ usually. We illustrate the running time of every critical subprocedure of both algorithms in Table 5. Our $k$-means step is faster, but we have to run an extra LP step. Therefore, the running time comparison between these two algorithms is complex. Generally speaking, LP takes more time than $k$-means, which means our Algorithm 1 usually runs slower than NIPS19. However, with the development of LP solvers, we can expect that the runtime of Algorithm 1 could be further reduced with more advanced LP solvers.

|  |  | Construct T | LP on T | k-means | LP on S | Rounding | Total |
|---|---|---|---|---|---|---|---|
| Bank | Algorithm1 | 0.01 | 2.4 | <0.01 | 1.23 | <0.01 | 3.78 |
|  | NIPS19 | / | / | 0.14 | 0.81 | <0.01 | 1.11 |
| Creditcard | Algorithm 1 | 0.01 | 4.06 | <0.01 | 2.27 | <0.01 | 6.51 |
|  | NIPS19 | / | / | 0.18 | 2.05 | <0.01 | 2.39 |
| Census1990 | Algorithm 1 | 0.01 | 7.51 | 0.02 | 5.19 | <0.01 | 12.99 |
|  | NIPS19 | / | / | 0.30 | 3.94 | <0.01 | 4.42 |
| Adult | Algorithm 1 | 0.01 | 4.14 | <0.01 | 1.80 | <0.01 | 6.12 |
|  | NIPS19 | / | / | 0.18 | 1.23 | <0.01 | 1.59 |
| Breastcancer | Algorithm 1 | 0.01 | 0.19 | <0.01 | 0.82 | <0.01 | 1.33 |
|  | NIPS19 | / | / | 0.10 | 0.22 | <0.01 | 0.45 |

Table 5: Running time (s) on non-strictly fair datasets

|  |  | Construct T | LP on T | k-means | LP on S | Rounding | Total |
|---|---|---|---|---|---|---|---|
| Moons | Algorithm 1 | 0.01 | 0.18 | <0.01 | 0.64 | <0.01 | 0.83 |
|  | NIPS19 | / | / | 0.07 | 0.70 | 0.01 | 0.78 |
|  | Algorithm 2 | / | / | <0.01 | / | / | 0.59 |
|  | ORL21 | / | / | 0.02 | / | / | 0.48 |
| Cluto | Algorithm 1 | 0.01 | 1.01 | <0.01 | 1.30 | <0.01 | 2.36 |
|  | NIPS19 | / | / | 0.07 | 1.54 | <0.01 | 1.66 |
|  | Algorithm 2 | / | / | < 0.01 | / | / | 0.56 |
|  | ORL 21 | / | / | 0.56 | / | / | 0.72 |
| Complex | Algorithm 1 | 0.01 | 1.08 | <0.01 | 0.61 | <0.01 | 1.71 |
|  | NIPS19 | / | / | 0.05 | 0.72 | <0.01 | 0.79 |
|  | Algorithm 2 | / | / | < 0.01 | / | / | 0.58 |
|  | ORL21 | / | / | 0.56 | / | / | 0.72 |
| Hypercube | Algorithm 1 | 0.01 | 5.71 | 0.01 | 4.40 | <0.01 | 10.27 |
|  | NIPS19 | / | / | 0.15 | 2.58 | <0.01 | 2.87 |
|  | Algorithm 2 | / | / | < 0.01 | / | / | 0.39 |
|  | ORL21 | / | / | 0.67 | / | / | 0.83 |

Table 6: Running time (s) on strictly fair datasets

**Discussion on the construction of $T$.**  According to Algorithm 1, $T$ should be an approximate centroid set (Matoušek, 2000). Thanks to the open-source project by (Kanungo et al., 2002), which provides an efficient implementation of the approximate centroid set, we used their algorithm as part of our procedure in our code. Kanungo et al. (2002) used a sampling technique, leading to a trade-off between performance and efficiency. In our experiment, we sampled 10% of points in the approximate centroid set as $T$. A higher sample rate yields better performance (lower cost) but longer running time.

Besides, an implicit benefit of the construction of $T$ is that it is irrelevant to the parameters $k$, $\alpha$, and $\beta$. So if we consider a real scenario that we need to repeatedly try different choices for these parameters (e.g., we may want to tune the value $k$ and select the most satisfying result), the step of constructing $T$ and performing linear programming on $T$ can be seen as preprocessing of datasets before the tuning. Namely, we just need to run this preprocessing one time, and consequently the amortized cost over the whole tuning procedure can be reduced significantly.

**Running time comparison on strictly fair datasets.**  For strictly fair datasets, we consider Algorithm 1, NIPS19, Algorithm 2, and ORL21 . Algorithm 2 has an advantage in efficiency in most datasets. The primary reason is that Algorithm 2 only calls the $k$-means algorithm once and does not need to solve the LP. As for ORL21, it needs to run $k$-means for each group and then choose the best one. As a result, ORL21 takes longer time than Algorithm 2, especially on the datasets with large number of groups.

### F.8    Experiments of Our Rounding Algorithm

In this section, we implement our rounding algorithm in Appendix C and compute the violation factor across different datasets and parameters. For convenience, we parameterize $\alpha_i$ and $\beta_i$ for the $i$-th group using a single parameter $\delta$. Specifically, we set $\beta_i = \frac{|P^{(i)}|(1-\delta)}{|P|}$ and $\alpha_i = \frac{|P^{(i)}|}{|P|(1-\delta)}$. Generally speaking, the smaller the $\delta$, the stricter the fairness constrains are. In Table 7 8 9, the violation introduced by our rounding algorithm is less than 1 in most of the cases and never exceeds 2, which aligns with our theoretical analysis.

| dataset | k=2 | 4 | 6 | 8 | 10 | 12 | 14 | 16 | 18 | 20 | 25 | 30 |
|---|---|---|---|---|---|---|---|---|---|---|---|---|
| Moons | 0 | 0 | 0 | 0 | 0 | 0 | 0 | 0 | 0 | 0 | 0 | 0 |
| Hypercube | 0 | 0 | 0 | 0 | 0 | 0 | 0 | 0 | 0 | 0 | 0 | 0 |
| Complex | 0.82 | 0.89 | 0.5 | 0.83 | 0.96 | 0.95 | 0.87 | 0.95 | 0.91 | 0.85 | 0.80 | 0.89 |
| Cluto | 0.80 | 0.86 | 0.72 | 1.01 | 1.04 | 0.94 | 1.0 | 1.02 | 0.90 | 0.90 | 1.1 | 0.9 |
| Biodeg | 0.05 | 0.66 | 0.65 | 0.63 | 0.64 | 0.62 | 0.63 | 0.68 | 0.77 | 0.79 | 0 | 0.01 |
| Breastcancer | 0.33 | 0.34 | 0.13 | 0.69 | 0.87 | 0.90 | 0.35 | 0.94 | 0.78 | 0.76 | 0.76 | 0.18 |

Table 7: Violation factor of our rounding algorithm with different $k$ ($\delta = 0$)

| dataset | k=2 | 4 | 6 | 8 | 10 | 12 | 14 | 16 | 18 | 20 | 25 | 30 |
|---|---|---|---|---|---|---|---|---|---|---|---|---|
| Moons | 0 | 0.3 | 0.35 | 0.40 | 0.30 | 0.40 | 0.70 | 0.5 | 0.35 | 0 | 0.20 | 0.40 |
| Hypercube | 0 | 0.94 | 0.98 | 0.94 | 0.83 | 0.95 | 0.85 | 0.91 | 0.80 | 0.88 | 1.02 | 0.83 |
| Complex | 0.67 | 0.98 | 0.66 | 0.87 | 0.88 | 0.97 | 0.76 | 0.77 | 0.89 | 0.97 | 0.67 | 1.03 |
| Cluto | 0.38 | 1.05 | 0.99 | 0.83 | 0.96 | 0.94 | 0.95 | 0.93 | 0.94 | 0.91 | 0.57 | 0.99 |
| Biodeg | 0 | 0.01 | 0.33 | 0.79 | 0.38 | 0.37 | 0.59 | 0.38 | 0.78 | 0.51 | 0.78 | 0.80 |
| Breastcancer | 0.18 | 0.23 | 0.40 | 0.23 | 0.39 | 0.89 | 0.53 | 0.33 | 0.47 | 0.51 | 0.34 | 0.68 |

Table 8: Violation factor of our rounding algorithm with different $k$ ($\delta = 0.1$)

| dataset | k=2 | 4 | 6 | 8 | 10 | 12 | 14 | 16 | 18 | 20 | 25 | 30 |
|---|---|---|---|---|---|---|---|---|---|---|---|---|
| Moons | 0 | 0.20 | 0.40 | 0.40 | 0.40 | 0.40 | 0.60 | 0.60 | 0.40 | 0.40 | 0.60 | 0.80 |
| Hypercube | 0 | 0.56 | 0.69 | 0.88 | 0.90 | 1.125 | 0.80 | 0.91 | 0.90 | 0.97 | 0.90 | 0.90 |
| Complex | 0.92 | 1.02 | 0.92 | 0.768 | 0.96 | 1.01 | 0.95 | 0.79 | 0.88 | 0.90 | 1.04 | 1.01 |
| Cluto | 0.85 | 0.90 | 0.90 | 0.88 | 0.83 | 1.024 | 0.85 | 0.86 | 0.90 | 0.88 | 0.96 | 1.00 |
| Biodeg | 0 | 0.50 | 0.56 | 0.39 | 0.51 | 0.69 | 0.19 | 0.57 | 0.56 | 0.64 | 0.75 | 0.65 |
| Breastcancer | 0 | 0.26 | 0.42 | 0.26 | 0.69 | 0.39 | 0.29 | 0.67 | 0.80 | 0.81 | 0.85 | 0.68 |

Table 9: Violation factor of our rounding algorithm with different $k$ ($\delta = 0.2$)

