# OpenReview forum: "Relax and Merge: A Simple Yet Effective Framework for Solving Fair $k$-Means and $k$-sparse Wasserstein Barycenter Problems"
_ICLR.cc/2025/Conference — ICLR 2025 Poster_

### Official Review · Reviewer_MEJb · 2024-11-01

**Soundness:** 3
**Presentation:** 3
**Contribution:** 3
**Rating:** 6
**Confidence:** 4

**Summary:**

The paper gives an approximation algorithm for the fair k-means algorithm, where, in addition to minimizing the sum of squared distances, the clusters should also satisfy certain minimum and maximum size constraints (stated as fractions of the data points). The paper gives (1+4c+O(\eps))-approximation algorithm with running time that has an exponential dependence on the dimension d and polynomial dependence on other parameters. The suggested algorithm first finds an \eps-centroid set T using Matousek's algorithm. This is followed by assigning points to centers in the set T by solving an LP that guarantees fairness. The size of T is exponential in dimension d. Since k centers need to be produced, the algorithm finds k centers by running a c-approximation algorithm for the vanilla k-means algorithm over points that are "moved" to their respective centers in T. Finally, integer assignments are found using ideas of Bera et al. (LP rounding), which causes minor violations in fairness conditions. This two-step process (clustering points using T followed by clustering T) is called Relax-and-Merge. Similar ideas have been used in the past (see, e.g., [A]), and the approximation analysis of the two-step process is similar to previous results (see Theorem 4 in [A]).

[A] https://theory.stanford.edu/~nmishra/Papers/clusteringDataStreamsTheoryPractice.pdf

Additional comment: I have reviewed an earlier version of the paper, and much of the review is from my earlier review. The current version improves the number of violations in the special case of mutually disjoint groups. This is shown by setting up an appropriate minimum-cost circulation problem.

**Strengths:**

- Fair clustering is a relevant problem, and the paper gives an approximation algorithm better than the known results in the Euclidean setting.

**Weaknesses:**

- The centroid set-based algorithm is specific to the Euclidean setting and cannot be extended to general metric spaces.
- The exponential dependence of the running time on d may be problematic for high-dimensional data.
- The techniques have mainly been borrowed from previous works. So, the theoretical contribution has limited novelty.
- PTAS for vanilla k-means either have exponential dependence on k or d. So, the ~ 5-approximation is at the cost of increasing the running time.

**Questions:**

Some questions can be found in the other part of review.

---

> ### Author Response · Authors · 2024-11-22
>
> ## "Weakness: The centroid set-based algorithm is specific to the Euclidean setting and cannot be extended to general metric spaces."
>
> Thanks for this question. It is true that the approximate centroid set can only be obtained in Euclidean space. In metric space, we usually assume that the potential facility set is given and its size is assumed to be polynomial in $n$. Under this assumption, our algorithm can be extended to general metric spaces. We use the given potential facility set as our candidate set, and we use the  triangle inequality, i.e., $dist(a,b) \le dist(a,c) + dist(b,c)$, to replace Proposition 1. We can also obtain $(1+2\rho)$-approximate solution for fair $k$-median and ($2+8\rho)$-approximation of fair $k$-means in metric space. The detailed proof is provided in Appendix E of our revised paper.
>
> Finally, we would like to emphasize that the major bottleneck of our approach for non-Euclidean space is the lack of the centroid set for them. If such a centroid set (or some similar structure) is available, we can immediately improve the approximation factor for those cases. A highlight of our current paper is that it brings a new perspective to fair clustering problems, which comes from this classical geometric structure. We hope that our work could inspire more improvements to fair clustering in future.
>
>
> ## "Weakness: The exponential dependence of the running time on d may be problematic for high-dimensional data." and "Weakness: PTAS for vanilla k-means either have exponential dependence on k or d. So, the ~ 5-approximation is at the cost of increasing the running time."
>
> We agree with the reviewer. In theory, the exponential dependence of the running time on $d$ arises from the inherent hardness of vanilla $k$-means. Even when $d$ is constant, the vanilla $k$-means problem remains NP-hard [1]. Hence, handling $k$-means in high-dimensional space is even more challenging. We agree that the PTAS algorithm is expensive. However, in practice, we can use $k$-means++, which is a popular method for solving the $k$-means problem and can be efficiently implemented. In our experiment, we demonstrated that using $k$-means++ can still achieve better performance than the baselines in most cases. In general, our algorithm does take higher running time for achieving the better clustering cost. In our comparison experiment, the running time of our algorithm is acceptable. Please refer to Appendix F.6 in our revised paper for a detailed discussion.
>
> [1] Mahajan M, Nimbhorkar P, Varadarajan K. The planar k-means problem is NP-hard[J]. Theoretical Computer Science, 2012, 442: 13-21.

---

> ### Author Response · Authors · 2024-11-23
>
> ## "Weakness : The techniques have mainly been borrowed from previous works. So, the theoretical contribution has limited novelty." and "Similar ideas have been used in the past (see, e.g., [A]),..."
>
> Thank you for providing reference [A]. We have carefully read the discussion on the $k$-median problem in that paper, including Theorem 4 mentioned by the reviewer. We agree that their framework is somewhat similar with our “relax and merge” framework at first glance, but we also would like to emphasize the important differences between them.
>
> * First, the algorithm of [A] is not easy to extend to address the fair $k$-means problem. [A] mainly focuses on the $k$-median problem in data streams. However, it is non-trivial to extend their method to solve the fair $k$-means problem. The major obstacle is how to maintain the **fairness** constraints inside their framework. In particular, a key challenge is to guarantee the partition and merge processes do not cause too much cumulative error regarding the fairness constraint.
>
>
> * Moreover, even if the method of [A] was able to be extended to our setting, the claimed approximation ratio of [A] would be higher than ours. In [A], the claimed ratio is $2c(1 + 2b) + 2b$, where $b$ and $c$ are the approximation ratios of $k$-clustering algorithms in two stages, while ours provides a $(1+4\rho)$ approximation factor. In addition, there are also several technical differences between the objectives of “k-means” and “k-median”.
>
> Nevertheless, we appreciate the reviewer for letting us know the work [A]. As we mentioned in our title, our two-step framework is "simple" yet "effective." We believe that this framework has potential to extend to other scenarios, such as in data streams, which is the major setting of [A]. We thank the reviewer for providing us with instructive material. We are glad to explore whether our framework can also be extended to a streaming fashion.
>
>
> Finally, we also want to mention that our contributions are not only the "relax and merge" approach, but also the establishment of a connection between the approximate centroid set and the fair $k$-means problem, which provides a new perspective for addressing fair clustering. We hope this new perspective could inspire more improvements to fair clustering in future. In addition, to address the unique “rounding” requirement of fair clustering (that not appears in ordinary k-clustering), our algorithms also combine several other key techniques.

---

> > ### Comment · Reviewer_MEJb · 2024-11-25
> >
> > Thank you for the response. I will maintain my initial score at this moment.

---

> > > ### Author Response · Authors · 2024-11-25
> > >
> > > Thank you again for your helpful comments.

---

### Official Review · Reviewer_9LbE · 2024-11-03

**Soundness:** 3
**Presentation:** 3
**Contribution:** 3
**Rating:** 8
**Confidence:** 5

**Summary:**

This paper considers the Euclidean fair $k$-means problem. In fair $k$-means, the input data set $G$ is partitioned into $m$ groups $G_1,..,G_m$ and $2m$ numbers $\alpha_1,..,\alpha_m,\beta_1,...\beta_m$ such that $\forall i\in[m], 0\leq \alpha_i\leq \beta_i\leq 1$. The task is to find $k$ clustering centers and partition $G$ into $k$ clusters so that for every $i\in [k],j\in [m]$ cluster $P_i$ must contain at least $\alpha_j|P_i|$  points and at most $\beta_j|P_i|$ points from group $G_j$.

The main contribution of this paper is two improved approximation algorithms for fair $k$-means. The first algorithm shows one can compute a $1+4\rho+\epsilon$ approximation with constant violation in fairness constraints by running upon an existing $\rho$-approximation algorithm for vanilla Euclidean $k$-means. This algorithm also relies on a centroid of size $O(n\epsilon^{-d}\log \epsilon^{-1})$ thus the running time is exponential on $d$. The second algorithm is designed for the strictly fair $k$-means, namely $\forall i, \alpha_i=\beta_i=\frac{1}{m}$. A $ 2+6\rho$ approximation without violation is obtained. Both algorithms have better approximation ratios than previous results. The authors also observe that the first algorithm can be extended to a related problem called $k$-sparse Wasserstein Barycenter.

Technically, authors observe that one can first compute an optimal assignment from the data set to a centroid set and then reduce the problem to the vanilla $k$-means by assigning each data point to its center in the centroid set. The idea is simple but effective. Experiments also show improvement over previous results.

Overall, I think this paper reaches the acceptance bar.

**Strengths:**

1. Improved approximation algorithms for an important problem.
2. The idea is simple but very useful.
3. Experiments show it is also promising for practice.

**Weaknesses:**

1. The running time of the first algorithm is exponential on $d$. Thus the result does not scale in high dimensions.
2. Eventually, the main contribution of this paper is a reduction via a centroid set. The performance relies on the centroid set construction algorithm and the approximation algorithm for vanilla $k$-means.

**Questions:**

1. Can one extend the results to the fair $k$-median?

---

> ### Author Response · Authors · 2024-11-22
>
> ## "Q1: Can one extend the results to the fair $k$-median?"
>
> Thanks for this question. Although the focus of our paper is mainly about k-means in Euclidean space, it is also deserved to provide the discussion for possible extensions. We need to discuss this issue separately for different scenarios.
>
> If we consider the $k$-median problem in Euclidean space, where facilities can lie in the ambient space, an immediate question is how to obtain a candidate set. In $k$-means, we have the method of [Matousek (2000)] to provide an "approximate centroid set". However, for $k$-median, as far as we know there is currently no method to construct such a candidate set to cover all potential optimal facilities. We would like to list it as an important open problem deserved to study in future.
>
> If we consider the $k$-median problem in general metric space or in Euclidean space but with a potential facility set whose size is polynomial and given, our framework can be extended to address this issue. We just directly use the whole facility set as our candidate set. As for the analysis, although we cannot use Proposition 1 to handle the $k$-median problem, we can use the triangle inequality instead. By doing so, we get a weaker upper bound of the approximation ratio compared with our result for Euclidean k-means -- $(1+2\rho)$ for the $k$-median problem, where $\rho$ is the approximation ratio for vanilla $k$-median. If the doubling dimension of metric space is fixed (this is a common assumption for some applications, since the doubling dimension measures data’s intrinsic dimension, which is usually low), then equipped with PTAS for vanilla $k$-median (such as the methods from [1, 2]), we can obtain a $3+\epsilon$-approximation for fair $k$-median. The detailed proof is provided in Appendix E of our revised paper.
>
> Finally, we would like to emphasize that the major bottleneck of our approach for handling k-median and non-Euclidean space is the lack of the centroid set for them. If such a centroid set (or some similar structure) is available, we can immediately improve the approximation factor for those cases. A highlight of our current paper is that it brings a new perspective to fair clustering problems, which comes from this classical geometric structure. We hope that our work could inspire more improvements to fair clustering in future.
>
> [1] Friggstad Z, Rezapour M, Salavatipour M R. Local search yields a PTAS for k-means in doubling metrics[J]. SIAM Journal on Computing, 2019, 48(2): 452-480.
>
> [2] Cohen-Addad V, Feldmann A E, Saulpic D. Near-linear time approximation schemes for clustering in doubling metrics[J]. Journal of the ACM (JACM), 2021, 68(6): 1-34.
>
> ## "Weakness 1: The running time of the first algorithm is exponential on d. Thus the result does not scale in high dimensions."
>
> We agree with the reviewer. In theory, the exponential dependence of the running time on $d$ arises from the inherent hardness of vanilla $k$-means. Even when $d$ is constant, the vanilla $k$-means problem remains NP-hard [1]. Hence, handling $k$-means in high-dimensional space is even more challenging. However, in practice, we can use $k$-means++, which is  a popular method for solving the $k$-means problem and can be efficiently implemented.  In our experiment, we demonstrated that using $k$-means++ can still achieve better performance than the baselines in most cases.
>
> [1] Mahajan M, Nimbhorkar P, Varadarajan K. The planar k-means problem is NP-hard[J]. Theoretical Computer Science, 2012, 442: 13-21.

---

> ### Comment · Reviewer_9LbE · 2024-11-25
>
> I have read the feedback and other reviews. I will keep my rating on this paper.

---

> > ### Author Response · Authors · 2024-11-26
> >
> > Thank you again for your review and comments.

---

### Official Review · Reviewer_GbWs · 2024-11-04

**Soundness:** 3
**Presentation:** 3
**Contribution:** 3
**Rating:** 6
**Confidence:** 4

**Summary:**

This paper studies the fair $k$-means clustering problem in Euclidean space, where the goal is to partition the given dataset into several groups such that each group should satisfy specific lower and upper bounds on the proportions of the assigned points. While the fair $k$-means clustering problem has been extensively studied in recent years, the existing approximation algorithms often incur at least a constant factor loss on the clustering quality guarantee in order to find good clustering centers for fair clustering instances that do not satisfy the “Voronoi properties”. To reduce the approximation loss, this paper introduces the notion of $\epsilon$-approximation centroid set, which can well approximate the locations for the optimal clustering centers under fairness constraints. Building on notion of the $\epsilon$-approximation centroid set, this paper proposes a Relax-and-Merge framework. The Relax-and-Merge framework first decomposes the given clustering instance into smaller partitions satisfying the fairness constraints (with more than $k$ partitions obtained) to construct a weighted instance. Then, by executing a weighted standard $k$-means algorithm on the weighted instance constructed, a solution satisfying the fairness constraints can be obtained  using linear programming rounding method, with better approximation guarantee on the clustering quality. Additionally, this paper shows that the proposed framework can also be extended to solve the $k$-sparse Wasserstein barycenter problem and strictly fair $k$-means problem. The experiments show that the proposed Relax-and-Merge framework achieves better clustering cost compared with other fair clustering algorithms, which further demonstrates the effectiveness of the proposed methods.

**Strengths:**

The strength of this paper can be summarized as follows.

1. Fair clustering is a topic that receives much attention in machine learning. If equipped with a PTAS for the $k$-means problem, the proposed algorithm can achieve the best approximation ratio of $5+O(\epsilon)$, which improves the current state-of-the-art approaches on clustering quality guarantees.

2. The proposed Relax-and-Merge framework is simple and effective. The technical analysis proposed to bound the loss when reducing the number of partitions to exactly $k$ is interesting.

3. This paper is well-written and the proofs are easy to follow.

4. The empirical evaluations demonstrate that the proposed framework achieves much better performances on clustering quality compared to the previous works.

**Weaknesses:**

The weakness of this paper can be summarized as follows.

1. The approximation guarantees of the proposed method rely on the approximation loss of the weighted algorithm used to reduce the number of partitions. Although many weighted algorithms (such as the local search algorithms) can achieve near-optimal clustering results in practice, the worst-case analysis rely on the use of a PTAS algorithm to achieve better approximation guarantees.

2. The proposed $\epsilon$-approximation centroid set is designed specifically to the Euclidean space. It is unclear that whether such methods can be extended to the general metric spaces.

**Questions:**

Q1: It seems that the proposed Relax-and-Merge framework is designed specifically for the Euclidean space. However, many fair algorithms can also be used to solve the case when data points are in a general metric space. Can the proposed method achieve better performances in general metric spaces compared with other fair clustering algorithms？

Q2: Since the $\epsilon$-approximation centroid set construction method does not rely on a specific constraint during the clustering process, whether the proposed framework can be extended to solve other fair clustering problems using $\epsilon$-approximation centroid set construction method?

Q3: In the experimental parts, the authors only compare the clustering quality of the proposed algorithms with other fair clustering algorithms. Can the authors give a detailed comparison between the running time of the proposed framework and other existing fair clustering algorithms?

Q4: The clustering quality of the proposed algorithm rely heavily on the weighted algorithm used to reduce the number of partitions. However, in the experimental parts, there is no detailed description of which weighted algorithm is chosen and how the weighted algorithm is executed. Which weighted $k$-means algorithm is used in the experiments for Algorithm 1? Did the authors choose a PTAS for the experiments?

---

> ### Author Response · Authors · 2024-11-22
>
> ## "Q1: It seems that the proposed Relax-and-Merge framework is designed specifically for the Euclidean space. However, many fair algorithms can also be used to solve the case when data points are in a general metric space. Can the proposed method achieve better performances in general metric spaces compared with other fair clustering algorithms?"
>
> Thanks for this question. Although the focus of our paper is mainly about k-means in Euclidean space, it is also deserved to provide the discussion for possible extensions.
>
> Our framework can be extended to a general metric space. We have added the proofs and discussions in Appendix F.6 of our revised paper. The proof idea is similar with Lemma 1. However, in a general metric space, we cannot use Proposition 1. Hence, we have to use the basic triangle inequality instead, which weakens the approximation ratio of our framework to ($1+2\rho)$ for fair $k$-median and $(2+8\rho)$ for fair $k$-means.
>
> Unfortunately, we should admit that the theoretical guarantees of our framework in general metric space are weaker than (Bera et al.). The primary reason is that in general metric space we do not have a proper candidate set similar to approximate centroid set in Euclidean space which holds Proposition 1. How to obtain a better candidate set is not only a potential future work of our framework but also an interesting open theoretical problem.  If such a centroid set (or some similar structure) is available, we can immediately improve the approximation factor for those cases. A highlight of our current paper is that it brings a new perspective to fair clustering problems, which comes from this classical geometric structure. We hope that our work could inspire more improvements to fair clustering in future.
>
>
> ## "Q2: Since the $\epsilon$-approximation centroid set construction method does not rely on a specific constraint during the clustering process, whether the proposed framework can be extended to solve other fair clustering problems using $\epsilon$-approximation centroid set construction method?"
>
> The answer is yes! Thank you for this question. The critical requirements of using our framework are twofold: (1) mergeable clusters; (2) optimal locating on centroids. (1) means that if you merge more than one feasible cluster into one, the merged cluster still remains feasible. (2) requires each optimal center to be the centroid of its clients.
>
> There are other types of constrained $k$-means satisfying (1) and (2), such as anonymous $k$-means [1, 3], which requires the number of points in every cluster to exceed some given threshold; and diversity $k$-means [2], which ensures the fraction of same-colored points in each cluster does not below some given parameter.
>
> In addition, we need to design a specific assignment algorithm for each constrained $k$-means. For the aformentioned two constrained problems, we can find assignment algorithms in [3].
>
> [1] Ahmadian S, Swamy C. Approximation Algorithms for Clustering Problems with Lower Bounds and Outliers[C]//43rd International Colloquium on Automata, Languages, and Programming (ICALP 2016). Schloss-Dagstuhl-Leibniz Zentrum für Informatik, 2016.
>
> [2] Li J, Yi K, Zhang Q. Clustering with diversity[C]//Automata, Languages and Programming: 37th International Colloquium, ICALP 2010, Bordeaux, France, July 6-10, 2010, Proceedings, Part I 37. Springer Berlin Heidelberg, 2010: 188-200.
>
> [3] Ding H, Xu J. A unified framework for clustering constrained data without locality property[J]. Algorithmica, 2020, 82(4): 808-852.
>
> ## "Q3: In the experimental parts, the authors only compare the clustering quality of the proposed algorithms with other fair clustering algorithms. Can the authors give a detailed comparison between the running time of the proposed framework and other existing fair clustering algorithms?"
>
> Thank you for this question. For the comparison of running time, please refer to our global response. We have also added Appendix F.6 to further discuss this issue.
>
> ## "Q4: The clustering quality of the proposed algorithm rely heavily on the weighted algorithm used to reduce the number of partitions. However, in the experimental parts, there is no detailed description of which weighted algorithm is chosen and how the weighted algorithm is executed. Which weighted k-means algorithm is used in the experiments for Algorithm 1? Did the authors choose a PTAS for the experiments?"
>
> We used $k$-means++ in our experimental section, as it is one of the most popular $k$-means algorithms in practice. In previous works, such as Bera et al., they also used $k$-means++ to solve the vanilla $k$-means problem. We have added a sentence in our experimental part to clarify this in our revised version. We did not choose PTAS for $k$-means as our $k$-means solver because it is too slow and complicated to implement. We used $k$-means++ as the $k$-means solver for our algorithm and all baselines.

---

> > ### Comment · Reviewer_GbWs · 2024-11-25
> > **Response to the Rebuttal**
> >
> > Thank you for the response. All of my questions have been addressed. Overall, this paper makes a valuable contribution by offering improved theoretical guarantees for the group fair clustering problem in Euclidean space, supported by experimental results that validate the theoretical guarantees. I will maintain my initial score at this moment.

---

> > > ### Author Response · Authors · 2024-11-25
> > >
> > > We  thank the reviewer again for the helpful comments.

---

### Official Review · Reviewer_9f2Z · 2024-11-05

**Soundness:** 3
**Presentation:** 3
**Contribution:** 3
**Rating:** 8
**Confidence:** 3

**Summary:**

$ (\alpha, \beta) $ -fair k-means clustering is a group-based notion of fairness where the number of points belonging to each protected 'group' (race, gender etc.) inside each cluster is bounded. The paper provides algorithms for $ (\alpha, \beta) $ -fair k-means clustering and strictly $ (\alpha, \beta) $ -fair k-means clustering that have better approximation ratios than existing works. This is achieved by considering the group-based fairness constraints before choosing the location of the centers rather than first choosing centers and then trying to assign points to centers according to fairness constraints.  The paper also gives approximation guarantees for $k$- Sparse Wasserstein Barycenter problem that tries to find an 'average' distribution with a support size of $k$, for given discrete $m$ distributions, by treating the problem as a special case of fair clustering.  The theoretical results are supported by empirical evaluations.

**Strengths:**

1) Fairness in clustering has gained a lot of attention in the past decade or so. For the case of $(\alpha, \beta) $ - fair clustering, the paper provides better approximation ratios than the existing works.

2) The paper for the most part is written very nicely and is not difficult to follow the central ideas. Sufficient Proof sketches/ details are provided in the main part of the paper. Although I could not check all the proofs in details, the paper appears technically sound.

3) The paper also shows the effectiveness of the proposed algorithms by empirical results on different datasets. Some of the empirical results can be better presented (see weaknesses) but overall, it is a very good mix of theory and practice and very suitable for the venue.

4) The high-level idea of considering fairness while choosing the location of centers and not after choosing centers might give insights to design better algorithms other fairness definitions also.

**Weaknesses:**

1) There is time complexity analysis of algorithm 1. However, there can be a comparison with the Bera et al. and Bohm et al. papers to get an idea of the tradeoff between approximation factor and time complexity. Empirical results should also result the time parameter. There are some experiments in the appendix, but they are just for your algorithm. Same goes for the violation ratio.

2) In terms of presentation, I think, it would be better to move some experiments to the main paper and one more proof to the appendices.

**Questions:**

1) Is it possible to obtain fair $k$- clustering without any fairness violation without the requirement of equally sized groups?

2) Can the method be extended to other clustering costs like $k$-median? It appears that proofs rely heavily on the cost being squared Euclidean distance.

3) Can you clarify how the solution to the k-sparse Wasserstein barycenter solution is guaranteed to be a distribution? Is the optimization slightly changed?

---

> ### Author Response · Authors · 2024-11-22
>
> ## "Q1: Is it possible to obtain fair k - clustering without any fairness violation without the requirement of equally sized groups?"
>
> Thank you for your thoughtful question. We can consider the following two cases separately (the latter one is more complicated than the first one):
>
> For the disjoint setting (where each point can belong to only one group), our new method declines the worst-case violation factor to 2. If we want to eliminate the violation completely,  as far as we know, previous work has achieved this by either (1) compromising the approximation ratio, such as [1] and [2], where they obtained an $O(\log k)$-approximation for fair $k$-median without violations, or (2) employing methods that are not strictly within polynomial time, such as the quasi-polynomial-time approximation scheme by [2], which also avoids violations. Achieving a (polynomial-time) constant-factor approximation without violations for $k$-clustering remains an open problem. We also add these references to our revised paper.
>
> For the overlapping setting (where a point can belong to multiple groups), to the best of our knowledge, no existing assignment algorithm completely avoids violations. Due to the hardness of the integer programming involved, eliminating the violations tends to introduce certain side effects, such as a larger approximation ratio or non-polynomial time complexity. We think this is an important open problem in the area of fair clustering, which is deserved to study in future.
>
> We will add more discussions and references in our paper about this topic after collecting all the final comments from the reviewers and ACs after this rebuttal phase.
>
>
> [1] Dai Z, Makarychev Y, Vakilian A. Fair representation clustering with several protected classes[C]//Proceedings of the 2022 ACM Conference on Fairness, Accountability, and Transparency. 2022: 814-823.
>
> [2] Wu D, Feng Q, Wang J. Approximation algorithms for fair k-median problem without fairness violation[J]. Theoretical Computer Science, 2024, 985: 114332.
>
> ## "Q2: Can the method be extended to other clustering costs like k-median? It appears that proofs rely heavily on the cost being squared Euclidean distance."
>
> Thanks for this question. Although the focus of our paper is mainly about k-means in Euclidean space, it is also deserved to provide the discussion for possible extensions. We need to discuss this issue separately for different scenarios.
>
> If we consider the $k$-median problem in Euclidean space, where facilities can lie in the ambient space, an immediate question is how to obtain a candidate set. In $k$-means, we have the method of [Matousek (2000)] to provide an "approximate centroid set". However, for $k$-median, as far as we know there is currently no method to construct such a candidate set to cover all potential optimal facilities. We would like to list it as an important open problem deserved to study in future.
>
> If we consider the $k$-median problem in general metric space or in Euclidean space but with a potential facility set whose size is polynomial and given, our framework can be extended to address this issue. We just directly use the whole facility set as our candidate set. As for the analysis, although we cannot use Proposition 1 to handle the $k$-median problem, we can use the triangle inequality instead. By doing so, we get a weaker upper bound of the approximation ratio compared with our result for Euclidean k-means -- $(1+2\rho)$ for the $k$-median problem, where $\rho$ is the approximation ratio for vanilla $k$-median. If the doubling dimension of metric space is fixed (this is a common assumption for some applications, since the doubling dimension measures data’s intrinsic dimension, which is usually low), then equipped with PTAS for vanilla $k$-median (such as the methods from [1,2]), we can obtain a $3+\epsilon$-approximation for fair $k$-median. The detailed proof is provided in Appendix E of our revised paper.
>
> Finally, we would like to emphasize that the major bottleneck of our approach for handling k-median and non-Euclidean space is the lack of the centroid set for them. If such a centroid set (or some similar structure) is available, we can immediately improve the approximation factor for those cases. A highlight of our current paper is that it brings a new perspective to fair clustering problems, which comes from this classical geometric structure. We hope that our work could inspire more improvements to fair clustering in future.
>
> [1] Friggstad Z, Rezapour M, Salavatipour M R. Local search yields a PTAS for k-means in doubling metrics[J]. SIAM Journal on Computing, 2019, 48(2): 452-480.
>
> [2] Cohen-Addad V, Feldmann A E, Saulpic D. Near-linear time approximation schemes for clustering in doubling metrics[J]. Journal of the ACM (JACM), 2021, 68(6): 1-34.

---

> ### Author Response · Authors · 2024-11-22
>
> ## "Q3: Can you clarify how the solution to the k-sparse Wasserstein barycenter solution is guaranteed to be a distribution? Is the optimization slightly changed?"
>
> After we run Algorithm 1, we obtain the support $S$ (the locations of centers) of the returned solution and the assignment matrix $\phi_S^*$. The key question is how to ensure that the summation of the weight of points in $S$ is equal to 1. The solution is simple: we just consider an arbitrary given distribution (or "group" in the context of fair $k$-means), e.g., $P^{(l)}$. For every facility $f$ in $S$, we define its weight $w(f) = \sum_{p\in P^{(l)}} \phi^*_S (p, f)$. This ensures that the total weight of $S$ must be equal to the total weight of $P^{(l)}$, which is 1 because $P^{(l)}$ is a distribution. The choice of $P^{(l)}$ can be arbitrary because, recall that $k$-sparse WB can be seen as a special fractional version of strictly fair $k$-means, meaning no matter which given distribution you choose, you will obtain the same weight distribution of $S$. The optimization will not change by setting the weight of $S$ because the weight of $S$ does not affect the cost.
>
> Thans for this question, and we also add the explaination to our revised paper Appendix D.
>
> ## "Weakness 1: There is time complexity analysis of algorithm 1. However, there can be a comparison with the Bera et al. and Bohm et al. papers to get an idea of the tradeoff between approximation factor and time complexity. Empirical results should also result the time parameter. There are some experiments in the appendix, but they are just for your algorithm. Same goes for the violation ratio."
>
> Thanks for your helpful suggestions.
>
> * For the comparison of running time, please refer to our global response. We have also added Appendix F.6 to further discuss this issue.
>
> * Regarding the violation factor, we have proven a 2-violation factor in theory, outperforming the baseline NIPS19 under the disjoint group setting (it is 7-violation for general overlapping setting, but in their paper they also claimed an improvement to 3-violation for disjoint setting). However, in practice, we find that NIPS19 (Bera et al.) always achieves near 0-violation factor, while ours tends to remain a small violation (as shown in Appendix F.7). The primary reason is that our rounding technique models the rounding procedure as an optimization (min-cost flow), that is, as long as the violation is within the bound 2, the algorithm will always search for the solution who has the cost as low as possible (i.e., reducing the cost has a higher priority than reducing the violation). This  results in a tendency that, even when our method has achieved a low-violation-factor solution at some point, it will continue searching for a lower-cost solution albeit suffering a larger violation (as long as it is bounded by 2). This tendency often leads our method to find lower-cost solutions but with slightly higer violation in practice, compared with the rounding method of NIPS19.
>
>   There are two simple ways to reduce the violation of our method in practice. The first solution is to add an early-stop rule to our rounding method that stops when all entries in the assignment matrix are integral. By doing so, we found that our rounding algorithm can also achieve near 0-violation in our datasets. The second solution is to run both rounding algorithms of NIPS19 and ours, then choose the solution that you prefer (note that the rounding step is very fast and takes only a minor part of the whole time, usually less than 1%, so running two rounding algorithms does not affect the total runtime too much). The rounding method of NIPS19 will stop immediately when all entries in the assignment matrix are integral, but it also gives up the potential to explore new solutions with lower costs. But our algorithm can address this shortcoming. Therefore, we can choose the one based on our priority, either focusing more on the cost or the violation.
>
> ## "Weakness 2: In terms of presentation, I think, it would be better to move some experiments to the main paper and one more proof to the appendices."
>
> Thank you for your suggestion. We will carefully modify our paper’s structure after collecting all the final comments from the reviewers and ACs after this rebuttal phase.

---

> > ### Comment · Reviewer_9f2Z · 2024-11-25
> > **Thanks for the rebuttal**
> >
> > I have read the author responses and other reviews. I believe this is a good paper and the rebuttal with additional experiments, discussion about $k$-median problem and other clarifications strengthen the paper. I encourage the authors to try and include at least some of it in the main part of the revised version. I am currently keeping my score.

---

> > > ### Author Response · Authors · 2024-11-25
> > >
> > > We would like to thank the reviewer again for the helpful comments and suggestions.

---

### Author Response · Authors · 2024-11-22

# Global Response

## Revised paper

We sincerely thank all the anonymous reviewers for their valuable and insightful comments. We have added some content to our revised paper according to the comments and questions. The changes are summarized below and marked in **blue** in the paper.
- *Page 1, Line 50-54*: We added more references about fair $k$-median algorithms.

 - *Page 9, Line 488*: We added a sentence stating that we use the $k$-means++ algorithm as our $k$-means solver.

- *Page 18, Appendix D*: We added a brief discussion on the k-sparse Wasserstein Barycenter problem for the sake of completeness.

 - *Appendix E*: We added detailed analysis on how to extend Algorithm 1 to $k$-median and $k$-means in general metric space.

 - *Appendix F.6*: We added an experiment comparing the running times of our Algorithm 1, Algorithm 2, and the baselines.

## Comparison of running time

For the comparison of running time, we added an experiment (the more detailed discussion is uploaded to Appendix F.6 of the revised paper). To shed some light, below we show an illlustrative example with $k=10$. We compare the running time of Algorithm 1, Algorithm 2 (for only strictly fair datasets), NIPS19 (Bera et al.), and ORL21 (Bohm et al., for only strictly fair datasets). The brief running time comparison is shown in the following table.

|                 | Bank | Creditcard | Census1990 | Adult | Breastcancer | Moons | Cluto | Complex | Hypercube |
| :-------------: | ---- | :--------: | ---------- | :---: | :----------: | :---: | ----- | :-----: | --------- |
| Algorithm 1 (s) | 3.78 |    6.51    | 14.95      | 6.12  |     1.33     | 0.83  | 2.36  |  1.71   | 10.27     |
|   NIPS19 (s)    | 1.11 |    2.39    | 4.42       | 1.59  |     0.45     | 0.78  | 1.66  |  0.78   | 2.87      |
| Algorithm 2 (s) | /    |     /      | /          |   /   |      /       | 0.59  | 0.56  |  0.58   | 0.39      |
|    ORL21 (s)    | /    |     /      | /          |   /   |      /       | 0.30  | 0.72  |  0.72   | 4.33      |

From the table, we conclude that for most strictly fair datasets, Algorithm 2 runs fastest. However, in general (non-strictly fair) datasets, our Algorithm 1 takes longer running time compared to NIPS1.

Here, we also provide some deeper insights for this phenomenon. Algorithm 1 and NIPS19 both have two important subprocedures: linear programming and the $k$-means algorithm. These two steps are the efficiency bottlenecks. Specifically, NIPS19 first runs the $k$-means algorithm (i.e., $k$-means++), and then  calls the LP solver once to compute the fractional assignment. A different part of our Algorithm 1 is that it calls the LP solver **twice**, once to compute the weights of candidate set $T$ and once to compute the fractional assignment, and calls the $k$-means algorithm once. In Algorithm 1, we only need to run $k$-means on $T$, which should be much smaller than the whole dataset, leading to less running time for the $k$-means subprocedure compared to NIPS19. However, the first call to the LP solver to compute the weight of $T$ consumes more time than the second call because $|T| > k$ usually. Generally speaking, LP takes more time than $k$-means, which means our Algorithm 1 usually runs slower than NIPS19. In fact, the first LP takes more than a half of overall time in most of the datasets according to the following table (the detailed comparison is shown in Table 5 and Table 6 of our revised paper).

|               | Bank | Creditcard | Census1990 | Adult | Cluto | Complex | Hypercube |
| :-----------: | ---- | :--------: | ---------- | :---: | ----- | :-----: | --------- |
| LP on $T$ (s) | 2.4  |    4.06    | 7.51       | 4.14  | 1.01  |  1.08   | 5.71      |
|  Overall (s)  | 3.78 |    6.51    | 12.99      | 6.12  | 2.36  |  1.71   | 10.27     |
|     Ratio     | 63%  |    62%     | 58%        |  68%  | 43%   |   63%   | 56%       |


Besides, we would like to point **an implicit benefit** of the construction of T is that it is irrelevant to the parameters $k$, $\alpha$, and $\beta$. So if we consider a real scenario that we need to repeatedly try different choices for these parameters (e.g., we may want to tune the value k and select the most satisfying result), the step of constructing T and performing linear programming on T can be seen as preprocessing on the data  before the tuning. Namely, we just need to run this preprocessing one time, and consequently the amortized cost over the whole tuning procedure can be reduced significantly.

As for strictly fair datasets, Algorithm 2 has an advantage in efficiency for most datasets.  The primary reason is that Algorithm 2 only calls the k-means algorithm once and does not need to solve the LP. As for ORL21, it needs to run k-means for each group and then choose the best one. As a result, ORL21 takes longer time than Algorithm 2, especially on the datasets with large number of groups.

---

### Meta-Review · Area_Chair_Eyjm · 2024-12-26

**Metareview:**

The notion of $(\alpha, \beta)$ fair clustering was introduced to consider $k$-means and other clustering algorithms with fairness constraints. The paper provides improved approximation algorithms for this problem and also the problem of $k$-sparse Wasserstein Barycenter. The results are interest and the paper has some algorithmic novelty; the results are also supported by empirical evidence.

**Additional Comments On Reviewer Discussion:**

The reviewers engaged well with the authors and all reviewers are at least mildly positive and two reviewers are strongly positive about the paper.

---

### Decision · Program_Chairs · 2025-01-22

Accept (Poster)